# Limits of Transformer Language Models on Learning to Compose Algorithms

**Jonathan Thomm**[1,2*]
jthomm@ethz.ch

**Giacomo Camposampiero**[1,2]
giacomo.camposampiero1@ibm.com

**Aleksandar Terzic**[1,2]
aleksandar.terzic1@ibm.com

**Michael Hersche**[1]
michael.hersche@ibm.com

**Bernhard Schölkopf**[2,3]
bs@tuebingen.mpg.de

**Abbas Rahimi**[1]
abr@zurich.ibm.com

[1]IBM Research – Zurich, [2]ETH Zurich, [3]MPI Tübingen

## Abstract

We analyze the capabilities of Transformer language models in learning compositional discrete tasks. To this end, we evaluate training LLaMA models and prompting GPT-4 and Gemini on four tasks demanding to learn a composition of several discrete sub-tasks. In particular, we measure how well these models can reuse primitives observable in the sub-tasks to learn the composition task. Our results indicate that compositional learning in state-of-the-art Transformer language models is highly sample inefficient: LLaMA requires more data samples than relearning all sub-tasks from scratch to learn the compositional task; in-context prompting with few samples is unreliable and fails at executing the sub-tasks or correcting the errors in multi-round code generation. Further, by leveraging complexity theory, we support these findings with a theoretical analysis focused on the sample inefficiency of gradient descent in memorizing feedforward models. We open source our code at `https://github.com/IBM/limitations-lm-algorithmic-compositional-learning`.

## 1 Introduction

While Large Language Models (LLMs) are known to perform well on natural language generation tasks [1, 2], they exhibit failures on reasoning [3, 4, 5, 6, 7, 8, 9, 10], mathematics [11, 12], causal inference [13, 14, 15], and algorithmic tasks [16, 17, 18]. Many interesting algorithmic tasks rely on function composition, which is an act of combining simple functions (e.g., primitive sub-tasks) to build more complicated ones. In this paper, we dive into the question of how sample-efficient Transformer-based [19] language models are when learning to compose as well as to decompose algorithmic procedures. We empirically approach the aforementioned question by analyzing the performance of Transformer language models on a set of compositional algorithmic tasks. Given that a Transformer language model has enough samples to learn all primitive sub-tasks, we define four hypotheses on how well it learns the composition task within the same training routine:

$\mathcal{H}_1$. A Transformer language model learns the task composition with a constant number of samples (only used in the in-context learning setting).

$\mathcal{H}_2$. A Transformer language model learns a compositional task with fewer samples than those required to learn its most difficult sub-task.

---

[*]Research conducted at IBM Research – Zurich.

38th Conference on Neural Information Processing Systems (NeurIPS 2024).

$\mathcal{H}_3$. A Transformer language model learns a compositional task with fewer samples than the sum of the samples needed to learn every sub-task.

$\mathcal{H}_4$. A Transformer language model needs more data samples than in $\mathcal{H}_3$.

This paper makes the following main contributions:

- In Section 3, we introduce a family of two new algorithmic tasks with a compositional structure that is well-suited for creating systematic sub-tasks and testing compositionality. We specifically design these synthetic tasks with independently observable sub-tasks such that the composition is easily inferable from the sub-tasks but harder to learn from scratch.

- In Section 4, we train LLaMA models [2] on this family of the tasks as well as two tasks investigated by Dziri et al. [16]. We ensure that all necessary sub-tasks are learned and test how efficiently the models can learn their compositional re-combinations. We show that training LLaMA models from scratch fails to compose all learned sub-tasks under hypotheses $\mathcal{H}_2$ and $\mathcal{H}_3$, making $\mathcal{H}_4$ the most plausible hypothesis for all four tasks. Furthermore, we propose a formal bound, showing that current supervised gradient descent training fails in compositional learning in the limit.

- In Section 5, we investigate GPT-4 and Gemini on all tasks and observe their failures to perform the tasks, or multi-round code generation, with task description and various chain-of-thought examples. This shows that in-context learning with few samples is unreliable and fails to compose knowledge from sub-tasks and that $\mathcal{H}_1$ does not hold for the investigated tasks.

## 2  Preliminaries

**Computational graph** Let $A$ be a deterministic algorithm and let $\mathcal{F}_A$ be a set of deterministic primitive operations that can be used by $A$ during execution. Given an input $x$, we define the computational graph for the algorithm $A$ as $G_{A(x)} = (V, E)$. $G_{A(x)}$ is a weakly connected, directed acyclic graph. The nodes $v \in V$ represent all intermediate variables' values during $A$'s execution, while the edges $e \in E$ represent the function arguments involved in the computation of their target nodes. One source node $s$ describes the input $x$, while the (single) sink node $t$ represents the output $A(x) = t$. For every non-source node $v$, $op(v) \in \mathcal{F}_A$ denotes the operation applied to compute $v$. Figure 1 shows an example with a toy input for one of the tasks (PEN) later introduced in Section 3.

**Sub-task definition** While Dziri et al. [16] define one sub-task for each operation occurring in the nodes of $G_{A(x)}$, we relax this constraint and also consider sub-tasks composed of multiple operations from $\mathcal{F}_A$. However, we require every operation in the set of primitives $\mathcal{F}_A$ to be *independently observable*. Given the set $\mathcal{S}$ of all the defined sub-task graphs, we say that an operation $f = op(v)$ is independently observable in a sub-task graph $G'_{A(x)}$ if either:

- $v$ is the only non-source node in $G'_{A(x)}$ (i.e., there is only one operation in the sub-graph).

- $v$ is not the only non-source node in $G'_{A(x)}$ and all other operations in the nodes of $G'_{A(x)}$ are independently observable in the other sub-task graphs in $\mathcal{S} \backslash \{G'_{A(x)}\}$.

By using this constraint on the definition of the sub-tasks, we aim to achieve a middle-ground between completely synthetic settings (where each primitive is presented in isolation) and real-world settings (where the primitives are often grouped and correlated).

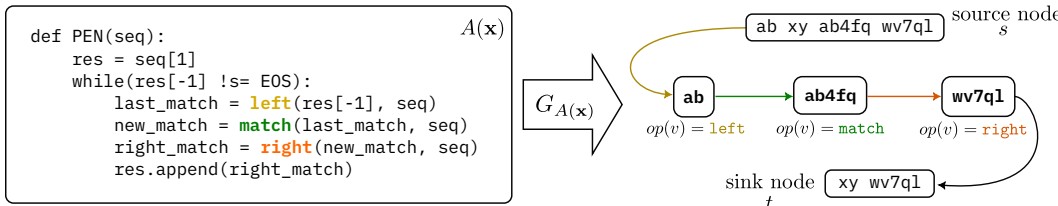

Figure 1: Translation of a compositional algorithmic task $A$, PEN (see Section 3 for details), into its corresponding compositional graph $G_{A(x)}$, for the input $x =$ "ab xy ab4fq wv7ql". The operations (edges) are color-matched with the respective operations in the pseudo-code of the algorithmic task $A(x)$.

# 3 Probing compositionality with algorithmic tasks

Pointer execution tasks are a family of algorithmic tasks initially proposed to benchmark the generalization capabilities of deep learning models [20]. They are designed to limit the number of confounders in the data and force the model to learn an algorithmic (general) solution, making solving the task with statistical shortcuts impossible. Furthermore, being algorithmic tasks, they are particularly suited for testing compositional learning, as they can be naturally decomposed into atomic operations. Hence, the introduction of this family of tasks would allow operating in a fully controlled environment, limiting the impact of exogenous factors on the empirical observations while having the possibility to stress-test compositionality.

Motivated by these premises, we introduce two novel pointer execution tasks for testing compositional learning, Pointer Execution's neighbor (PEN) and Pointer Execution Reverse Multicount (PERM). To ensure the validity of our empirical evaluation, we then further extend our experiments to two well-established tasks for testing compositionality in Transformer-based language models, Highest Subsequence Sum (HSS) and Multiplication (MUL) [16]. We decompose each task in a set of sub-tasks $\mathcal{S}$, such that every primitive operation of the task is independently observable in at least one sub-task. A more detailed exposition of the properties of $\mathcal{S}$ is included in Appendix B.1.

## 3.1 Pointer execution's neighbor (PEN)

We introduce the Pointer Execution's neighbor (PEN) task, where the goal is to jump between different words in the sequence according to a matching criterion while outputting the right neighbors of the matched words. This task is inspired by C-PVR, a task recently introduced by Abnar et al. [17] and itself based on the Pointer Value Retrieval task [20]. A sketch of the task is shown in Figure 2 (left). We identify three primitive operations to be `left` (get the green left neighbor), `match` (get the matching green word), and `right` (get the right yellow neighbor). We, therefore, split PEN into three sub-tasks that guarantee independent observability for each primitive: *copy* (Cpy), where the solver has to copy an input sequence of words (making the `right` primitive observable); *reverse copy* (RCpy), where the solver has to copy an input sequence of words in the reversed order (i.e. the last word first, making the `left` primitive observable); *Pointer Execution* (PE), where the solver has to match words in a sequence (making the `match` primitive observable). Additionally, we define the sub-task *Pointer Execution Verbose* (PEV) to facilitate the learning of the task. PEV requires solving the same problem as PEN with the addition of outputting both the matching words and their neighbors, making it less abstract (see Figure 2). More details on PEN can be found in Appendix B.2.

To make the task more challenging, we add "attention traps" to the input sequences. These traps add spurious matches between neighboring (yellow) tokens (two spurious matches per yellow neighbor), such that the model could be tricked into matching the wrong sequence of tokens (yellow instead of green). An example of this kind of trapping mechanism is included in Figure 2 (left column), and in a more explicit visualization in Figure B.5. The traps are not added to the main (green) tokens, where the sequence of matches is always deterministic.

## 3.2 Pointer execution reverse multicount (PERM)

We introduce the Pointer Execution Reverse Multicount (PERM) task. This task is conceptually similar to the PEN task. However, instead of matching forward and predicting the current word (or its neighbor), the solver has to output the *reversed* matched sequence. To increase the difficulty of the task, we enrich it with additional operations on the indices. In particular, for each element in the matched sequence, the solver needs to collect the number of matches and the number of left matches up to that point, multiply them, and output the result. A visualization of the task is presented in Figure 2. We omit attention traps because there are no longer neighbor tokens in the input sequence.

We identify the primitive operations of the PERM task to be `match` (common to PEN), `reverse` (reverse a sequence), and `multicount` (indexes operations, which includes counting the number of matches and left matches, as well as multiplying the two counts together). The primitives `left` and `right` are no longer needed, as we drop the concept of neighbors in this task. To make every primitive independently observable and learn PERM, we formulate three sub-tasks: *Pointer Execution* (PE), inherited from PEN; *Pointer Execution Reverse* (PER), where the solver has to match a sequence and reverse it, combining the primitives `match` and `reverse` (making the latter observable); *Pointer*

*Execution Multicount* (PEM), where the model has to match the sequence and compute the index operations, combining the primitives `match` and `multicount` (making the latter observable).

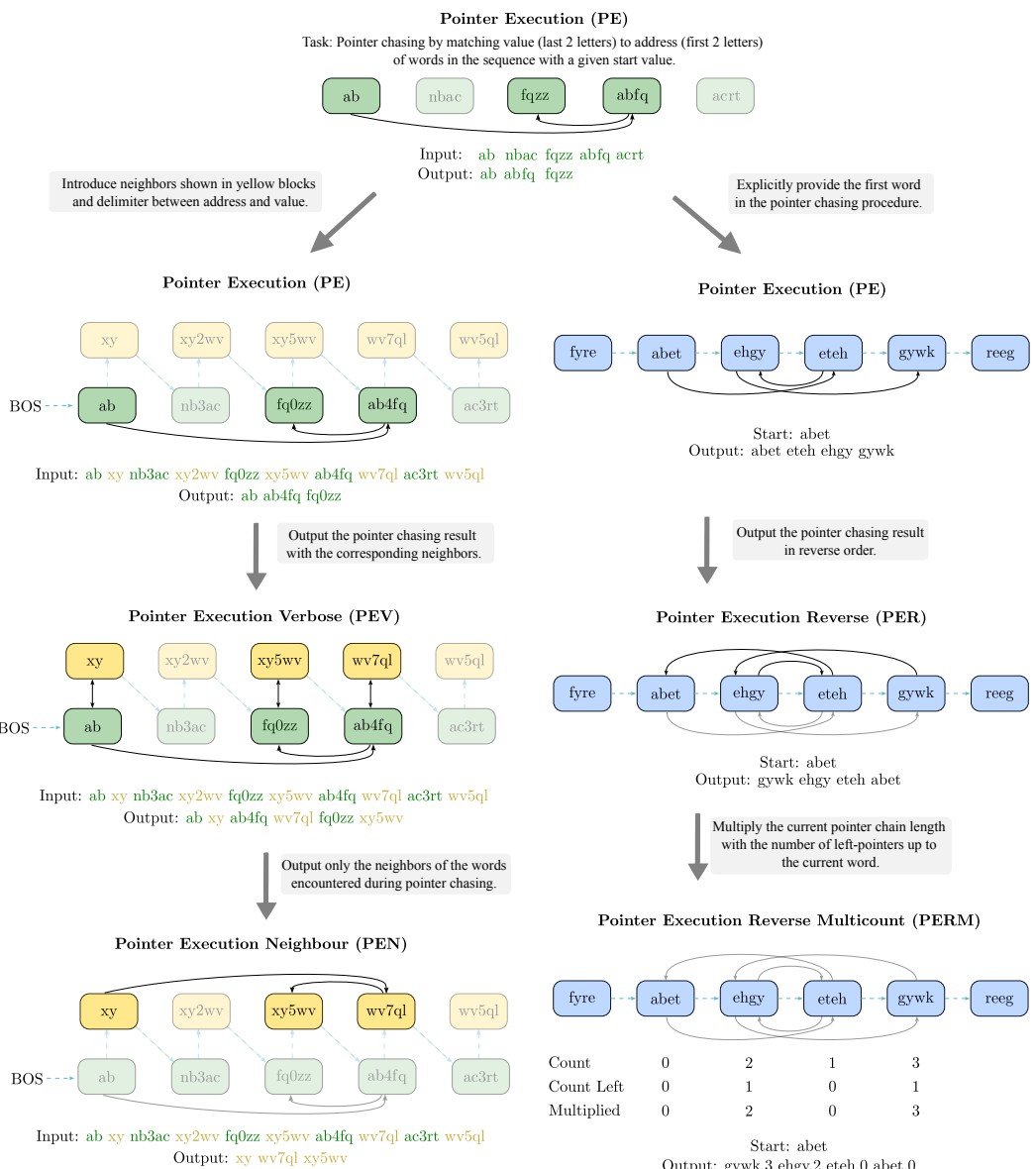

Figure 2: **Introduced compositional algorithmic tasks. Left:** The Pointer Execution (PE)'s neighbor (PEN), together with the Pointer Execution (PE) and Pointer Execution Verbose (PEV) sub-tasks. Starting left, the output is obtained by matching words and predicting the current word (in PE) or its neighbor (in PEV and PEN). Our matching criterion is that the two end characters of the current word are equal to the first two characters of the matched word. By ensuring that there are no ambiguities in the input string, an attention mechanism can find the match by retrieving the last two characters of the word and matching it with the (unique) word that starts with them. **Right:** The Pointer Execution Reverse Multicount (PERM), together with the Pointer Execution (PE) and Pointer Execution Reverse (PER) sub-tasks. PERM first outputs the last word in the matching sequence and then goes backward. The number in the answer for each word is the count of matches times the count of left matches (i.e., arrow to the left in the forward matching sequence).

### 3.3 Highest subsequence sum (HSS)

Given a sequence of numbers, the Highest Subsequence Sum (HSS) task [16] consists in finding the highest sum of a number subsequence where no two numbers are neighbors. For this task, there exists a simple linear-time, constant-space dynamic programming (DP) solution which we use to generate sub-tasks. The dynamic programming recurrence is

$$dp(i + 1) = \max(dp(i - 1) + \text{number}_{i+1}), dp(i)), \tag{1}$$

where $\text{number}_{i+1}$ is the $i + 1$-th number in the input sequence. The final answer is $dp(n)$ with $n$ being the length of the input sequence. We identify one fundamental primitive used in this task inspired by this formulation of the problem, `dp_step`, presented in Equation 1. Hence, we define a single sub-task, the Subsequence Sum Execution (SSE) sub-task, to make the primitive `dp_step` observable. In this sub-task, the solver is required to execute the DP recurrence, explicitly computing $dp(i)$ for the position $i$ in the output (including if the new number was taken or not).

### 3.4 Multiplication (MUL)

The Multiplication (MUL) task [16] involves multiplying two multi-digit numbers in base 10. It is often used to assess the symbolic and compositional capabilities of LLMs [21, 22]. Dziri et al. [16] use this task to find that LLMs do not generalize well to out-of-domain computation graph depth and width. We identify two main primitive operations: `digit_mul` (digit multiplication between a number in base 10 and a digit) and `add` (addition between numbers). We then formulate two corresponding sub-tasks to guarantee that each one of these operations is observable: digit-multiplication (MUL), where the solver has to solve multiplications between a number in base 10 and a digit (making `digit_mul` observable) and addition (ADD), where the solver has to add numbers in base 10 (making `add` observable). We do not explicitly train on shifting the numbers with zeros before adding them up.

## 4 Sample efficiency on compositional learning

### 4.1 Training LLaMA models for testing compositionality

In this section, we investigate compositional learning on the tasks presented in Section 3. For each one of them, we encode its multiple sub-tasks by adding unique identifiers at the beginning of the samples. We then train a LLaMA model from scratch on all sub-tasks concurrently (sampling from them uniformly at random). We use a character tokenizer and apply cross-entropy loss on all tokens, including both the input and the output sequences, as is usual in autoregressive language modeling. Computing the loss only on the answer part, as done in other works [23, 24], led to worse results in our experiments. We consider an algorithmic task to be learned only when (almost) perfect in-distribution accuracy is achieved. Any solution resulting in lower accuracy is, on the other hand, considered wrong, as it fails to fully capture the operations of the algorithm underlying the task. We conducted additional ablations on the model architecture (e.g., a LLaMa architectural variation based on Universal Transformer [17]), included in Appendix C.5. However, the unsatisfactory results deterred us from conducting further systematic analysis on it.

**Learning the PEN task.** We train LLaMA with all four sub-tasks (Cpy, RCpy, PE, PEV) and PEN. As shown in Figure 3 (detailed numbers are in Table C.2), the model successfully learns all the sub-tasks containing the primitives needed for PEN. However, it fails to properly compose them for solving the task under hypotheses $\mathcal{H}_2$ (no. of samples needed for the most difficult sub-task) and $\mathcal{H}_3$ (no. of samples needed for relearning all sub-tasks). Increasing the sample size to 1000 K (roughly $10\times$ more compared to the sum of the samples needed to learn each sub-task independently) allows the model to learn the PEN task. This makes hypothesis $\mathcal{H}_4$ (a Transformer language model needs more data samples than the sum of samples needed for re-learning all sub-tasks separately) the most plausible.

Furthermore, by training the model on PEN only (without the sub-tasks), we observe that LLaMA obtains no significant benefit from learning the sub-tasks (see w/o sub-tasks vs. w/ sub-tasks in Figure 3). Based on these results, we speculate that the sub-tasks are not properly reused, and the tasks are rather learned every time from scratch. We interpret this as evidence that the model is not learning the task compositionally.

To set those results into context: our dataset of 1000 K samples overall has roughly $6\times$ more tokens than the number of words a typical 13-year-old would have heard in their life [25]. Furthermore, we can train a discrete hill-climbing learning algorithm containing 121 discrete parameters to perfectly learn the PEN task from a single sample, given the sub-tasks as primitives (see Appendix C.2).

We also propose an ablation with a smaller model (28 M instead of 150 M parameters, "aux. loss" in Figure 3, and more detailed Table C.3). We add two additional prediction heads after a fraction of $1/2$ and $3/4$ of the layers, respectively, and train them using auxiliary losses. The first loss is used to train the first head to predict, at step $i$, the $i$-th (yellow) element of the answer. The second loss is used to train the second head to predict the (green) element to the left of the one predicted by the first head. The auxiliary losses model can learn PEN with 300 K samples, while the bigger LLaMA (150 M) obtains 0% accuracy with the same dataset size. However, this result also falls into the $\mathcal{H}_4$ hypothesis.

**Learning the PERM task.** We train LLaMA models on the PERM task together with its sub-tasks (PE, PER, and PEM). The training setting used to train the model on this task is identical to the one used for PEN. As reported in Figure 3 (and with more detail in Table C.4), LLaMA can learn with perfect accuracy all the individual sub-tasks but fails to learn their composition (PERM) within hypotheses $\mathcal{H}_2$, and $\mathcal{H}_3$. To achieve perfect accuracy, we need to scale up the dataset size by almost one order of magnitude compared to the $\mathcal{H}_3$ setting (the required number of training samples is equal to the sum of the training samples of all the sub-task datasets). This provides evidence that hypothesis $\mathcal{H}_4$ is the most plausible for PERM too. As for PEN, we study the impact of providing the sub-tasks together with the main task during training. Comparing the performance of LLaMA when given the sub-tasks and not (cf. Figure 3 top right), we observe that LLaMA seems to not benefit significantly from the sub-tasks. We speculate that also in this case this is a sign that the model is not re-using the primitives learned from the sub-tasks to solve the compositional task.

**Learning the HSS task and MUL tasks.** We investigate HSS and MUL (related to [16]). Figure 3 (bottom row) visualizes the results. As with our PEN and PERM tasks above, we observe that LLaMA does not exhibit strong compositional learning and learns under hypothesis $\mathcal{H}_4$. Also for

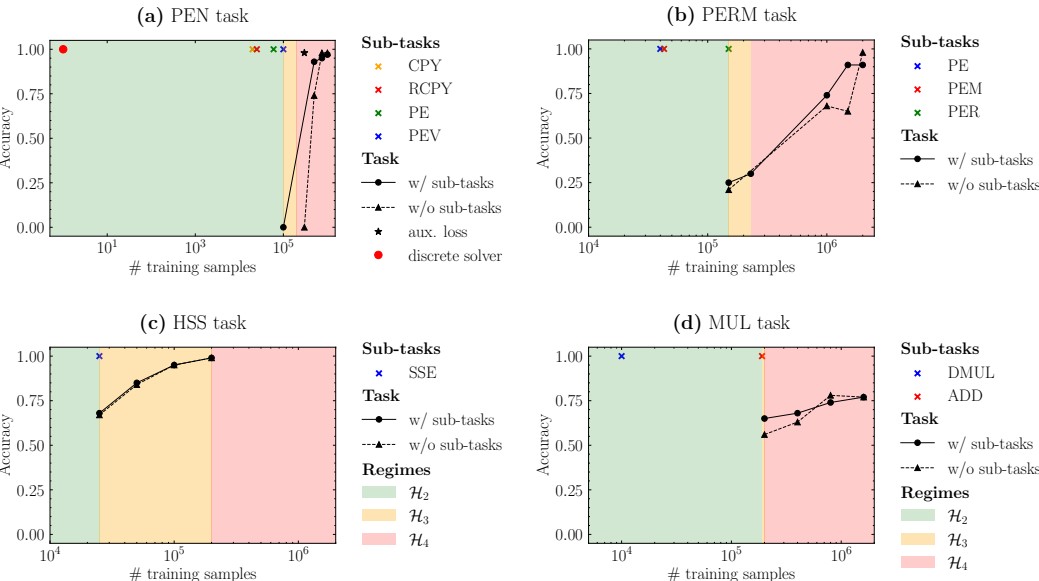

Figure 3: **Accuracy of LLaMA models on PEN, PERM, HSS, and MUL, and their respective sub-tasks.** While LLaMa achieves perfect accuracy on all the individual sub-tasks, it needs much larger amounts of training data to learn their composition. This observation makes hypothesis $\mathcal{H}_2$ (learning the composition requires less samples than the hardest sub-task, green) and $\mathcal{H}_3$ (learning the composition requires less samples than the sum of the sub-tasks, yellow) impossible to achieve on every task, where LLaMa seems to always fall into $\mathcal{H}_4$ (learning the composition requires more samples than the sum of the sub-tasks, red). Moreover, training together with the sub-tasks does not seem to perform sensibly better than training only on the main task (i.e., w/o sub-tasks).

these tasks, it is possible to see that learning the sub-tasks along with the composition task did not significantly improve the performance.

## 4.2 Testing decomposition by learning sub-tasks from a pre-trained LLaMA model

We additionally evaluate LLaMA's abilities on decompositionality, i.e., learning a sub-task like PEV after being trained only on the full compositional task (PEN). We observe that, when learning the decomposition task, the models benefit from being previously pre-trained on the composition task (see Table C.10). However, to get maximum accuracy on decomposing PEV from PEN, we need between 25 K and 50 K samples to fine-tune a PEN-pretrained model. This is a significant fraction of the data to learn the PEV task from scratch, making $\mathcal{H}_1$ also implausible for decompositionality on the PEN task. Similar observations hold for decomposing PE from PERM (i.e., a PERM-pretrained model fine-tuned on the PE task; see Table C.11). We speculate that this might be an indication that the model does learn operations relevant for the sub-tasks when trained only on the composite task (since learning the decomposition gets easier), and the learned solution might be to some extent compositional, even if the learning process is not compositional (and very inefficient).

## 4.3 Asymptotic limitations on compositional learning

In general, compositional learning can be split into two skills: (1) identifying and learning the primitives from the sub-tasks, and (2) assembling the learned primitives in the correct way to construct the solution to the composition task. In Appendix A we focus on (2) and enrich our empirical results with a theoretical perspective; a gist of it is included in the following paragraphs.

**Pre-training LLMs is sample inefficient in compositional learning.** We prove that asymptotically, meaning from an unknown size onwards, feedforward models on gradient descent are data inefficient on combinatorial (including compositional) problems. We use a reduction over the required asymptotic compute time to prove the following result.

**Theorem 4.1** (Informal Theorem A.7)**.** *There exist (many) series of train-test datasets, such that feedforward models on gradient descent that memorize samples will need $O(n^k)$ times more data than an optimal learner. Otherwise, they will not generalize to the test set. $k$ is any positive number and $n$ is the index in the series.*

**Proof sketch.** Many combinatorial problems are assumed to be hard to solve. $NP$ hard problems provide an extreme case, where under the $P \neq NP$ assumption the solution algorithm requires more than polynomial time. By constructing a train and test dataset where the problem instance of a combinatorial problem (e.g., SAT) is in the training set and the solution can easily be extracted from the labels of the test set, we can make the following reduction: if the learning algorithm finishes in polynomial time, then it cannot answer the test samples correctly, as this would build an algorithm which can solve the problem faster than possible. If our learning algorithm, however, tends to memorize the training set fast and, after memorization, training does not improve model test performance anymore, we can conclude that many samples will be needed to be able to generalize to the test set. At the same time, we can simulate a slow but data-efficient algorithm that extracts the problem instance from the training data, computes the answer, and naturally generalizes it to the test set. The same reduction applies to non-$NP$-hard problems, in which case the $k$ in Theorem 4.1 becomes bounded. While the classical reduction requires an out-of-distribution test set, the reduction can be made in the in-distribution case as well using hardness assumptions from public key cryptography (shown in Theorem A.9).

## 5 Compositional learning via in-context prompting

In this section, we investigate the performance of pre-trained LLMs, GPT-4 and Gemini-Pro, on the PEN and PERM tasks with various prompting methods. Concretely, we operate under the assumption that the models have learned to execute a sufficient set of operations during pre-training, and we test whether these latent abilities can be utilized to accurately solve the tasks given prompts of various complexitites. We experiment with a wide range of prompts, ranging from simple few-shot examples to state-of-the-art methods [26, 27, 28]. Besides the task test accuracy, we also report two partial correctness metrics, *match accuracy* and *termination accuracy*. Match accuracy measures how many steps in the output were a correct `left-match-right` step, regardless of the last word being correct.

Termination accuracy checks if the last output word's left neighbor in the sequence matches the input sequence. We include ablations on possible confounding factors (such as tokenization), different task definitions, and number of in-context examples in Appendix D.

## 5.1 GPT-4 and Gemini prompting on PEN and PERM

**Prompting to solve PEN.** We first provide the models with simple few-shot prompts demonstrating eight input-output examples for the task. The exact format is shown in Figure D.16. We find that this is not sufficient to obtain non-zero task accuracy. We then prepend a natural language description of the task to the few-shot prompt as shown in Figure D.17, but we find that this does not change the scores in any significant way. We proceed further by introducing Chain-of-Thought (CoT) prompting [26]. The prompt now consists of a high-level description of the task, eight input-output examples, and a request for the model to perform the task step-by-step (Figure D.18). With this prompting method, the models still exhibit zero task accuracy. We perform a similar experiment using a more recent and advanced CoT query, analogical CoT [28], which prompts the model to generate its own few-shot examples by asking it to recall several examples of similar algorithmic problems it has been exposed to during training (Figure D.19). This still does not help us achieve non-zero task accuracy.

We then provide more explicit guidance in the few-shot examples using two different approaches. In the first approach (Few-shot CoT, Figure D.20), the few-shot examples demonstrate each step in the pointer matching procedure alongside with the corresponding outputs (right neighbors) at each step. In the second approach (Sub-task CoT, Figure D.21), the few-shot examples consist of two phases: first, we match the entire sequence given an initial input, and then we output the right neighbors of the elements of the previously matched sequence. While none of the two approaches achieve non-zero task accuracy, sub-task CoT prompts induce a higher match accuracy in the Gemini-Pro model, reaching 33%.

We experiment with one further ablation, in which we remove the additional attention traps in the yellow tokens (see Appendix B.2), such that every yellow token only matches once. Although this

| Task | Few-shot | Description | CoT | Few-shot CoT | Sub-task CoT | Traps Removed | Code Interpreter | Analogical CoT | GPT-4 Termination Acc. | GPT-4 Match Acc. | GPT-4 Task Acc. | Gemini-Pro Termination Acc. | Gemini-Pro Match Acc. | Gemini-Pro Task Acc. |
|---|---|---|---|---|---|---|---|---|---|---|---|---|---|---|
| | | | | | Prompt Setting | | | | | GPT-4 | | | Gemini-Pro | |
| PEN | ✓ | | | | | | | | 0.12 | 0.04 | 0.00 | 0.05 | 0.03 | 0.00 |
| | ✓ | ✓ | | | | | | | 0.11 | 0.04 | 0.00 | 0.09 | 0.03 | 0.00 |
| | ✓ | ✓ | ✓ | | | | | | 0.08 | 0.14 | 0.00 | 0.05 | 0.01 | 0.00 |
| | ✓ | ✓ | ✓ | ✓ | n.a. | | | | 0.16 | 0.23 | 0.00 | 0.19 | 0.12 | 0.00 |
| | ✓ | ✓ | ✓ | n.a. | ✓ | | | | 0.20 | 0.14 | 0.00 | 0.14 | **0.33** | 0.00 |
| | ✓ | ✓ | ✓ | ✓ | n.a. | ✓ | | | 0.30 | 0.39 | 0.00 | **0.20** | 0.08 | 0.00 |
| | 1 | ✓ | ✓ | n.a. | n.a. | | ✓ | | **0.31** | **0.41** | **0.19** | - | - | - |
| | n.a. | ✓ | n.a. | n.a. | n.a. | n.a. | | ✓ | 0.06 | 0.16 | 0.00 | 0.10 | 0.00 | 0.00 |
| PERM | ✓ | | | | | n.a. | | | 0.12 | 0.37 | 0.06 | 0.08 | 0.04 | 0.04 |
| | ✓ | ✓ | | | | n.a. | | | 0.14 | 0.45 | 0.00 | 0.10 | 0.10 | 0.02 |
| | ✓ | ✓ | ✓ | | | n.a. | | | 0.38 | 0.74 | 0.08 | **0.28** | 0.12 | 0.02 |
| | ✓ | ✓ | ✓ | ✓ | n.a. | n.a. | | | **0.80** | **0.96** | 0.14 | 0.06 | **0.82** | 0.00 |
| | ✓ | ✓ | ✓ | n.a. | ✓ | n.a. | | | 0.62 | 0.95 | **0.42** | 0.19 | 0.77 | **0.09** |
| | 1 | ✓ | ✓ | n.a. | n.a. | n.a. | ✓ | | 0.67 | 0.42 | 0.13 | - | - | - |
| | n.a. | ✓ | n.a. | n.a. | n.a. | n.a | n.a. | ✓ | 0.16 | 0.41 | 0.02 | 0.16 | 0.06 | 0.00 |

Table 1: **Results of GPT-4 and Gemini-Pro with various prompt settings** (including chain-of-thought (CoT), analogical CoT [28], and code interpreter as a multi-round prompting technique [27]). Despite providing strong hints, few-shot CoT, task description, and without double traps (single only), both GPT-4 and Gemini fail on PEN (0% task accuracy). On PERM, the trend is similar and the main obstacle is multi-counting correctly (cf. the high match accuracy in the last two rows). GPT-4 reaches 42% task accuracy with a hand-crafted sub-task CoT. Gemini-Pro reaches to 9% in this setting.

does not help GPT-4 or Gemini to obtain a correct answer (still 0% task accuracy), it more often chooses the correct word to match within its answer sequence, reaching 41% match accuracy. This showcases how susceptible GPT-4 is to adversarial or unfortunate task-irrelevant correlations, also within chain of thought.

We also ablate on the newly available `o1-preview` model from OpenAI. While it cannot infer the task only from examples, it reaches 70% accuracy given a description of how to perform the task and CoT; see Appendix D.4 for more details.

**Prompting to solve PERM.** The same progression of prompt refinements is also applied for PERM, and the results are consistent with the observations on the PEN task. The results are shown in Table 1. When asked for CoT, GPT-4 does not find a good way to calculate the numbers for each word and tries to determine directly whether a match is to the left or not, something it seemingly has trouble doing. The best performing method is "Sub-task CoT" (Figure D.22). In this method, we begin by explicitly enumerating all of the words. With this, a left match can be determined by comparing two numbers. We conclude the few-shot examples with an explicit computation of the multicount numbers. This format increases the GPT-4 task accuracy to 42%, which is significantly higher (by 29%) than all of the other results. However, this setting deviates from identifying and leveraging the compositional structure of the tasks, since it rather corresponds to executing a hand-crafted algorithm. Based on the presented results, we can conclude that on the investigated example compositions, in-context learning with few samples is unreliable and fails to compose knowledge from sub-tasks and solve the main compositional task.

## 5.2 GPT-4 code interpreter on PEN and PERM

We experiment with asking GPT-4 to write code, which is a suitable prompting method for our algorithmic tasks. We use the official multi-round code interpreter of GPT-4's assistant API. While it enables GPT-4 to sometimes output a correct solution, it is not systematic and often fails at finding its errors. An example prompt for PEN is shown in Figure D.23.

To get a deeper understanding of how the code generation of GPT-4 fails, we investigate 20 random answers from our results of the multi-round code interpreter in Table 1. We find that the model always outputs executable code and copies the task sequence correctly. While 10% of the answers were correct, 65% of the answers were a wrong answer sequence, and in the remaining 24%, the model tried 10 attempts of code programs and messages in between before being killed. In 35% of the cases, the model initially produced code which results in an infinite loop when executed on the sample. In those cases, the model can fix this problem once (13%), three times (38%) it tries to correct itself until shutdown, and three times (38%) it catches the problem but still outputs a wrong answer. Once it catches the error and gives a correct answer (13%). Further, it is interesting that the model never leverages the example question+answer given (it could verify its solution code with that before submitting an answer).

## 6 Related work

Dziri et al. [16] and others [9, 29, 30] investigate limitations of Transformer language models on compositionality with prompting and finetuning. Differently from them, we train state-of-the-art language models from scratch and analyze how well the models can reuse and reorder sub-tasks learned. This gives new insights into compositional learning and provides a view into some limitations of current language model pre-training. Similarly, Razeghi et al. [31] find a correlation between performance and the frequency of how often task instances occur in the training data.

Apart from the tasks we use in this work, there exist many benchmarks on compositional learning for language models [3, 5, 8, 11, 21, 32, 33, 34, 35]. However, compared to them, our training setup and synthetic tasks specifically target compositional learning abilities subject to sample efficiency.

Many works tackled the problem of improving models' compositionality already [9, 36, 37, 38, 39]. However, their focus is mostly on the improvement of the model's behavior in specific benchmarks, while model-inherent systematic compositional learning remains extremely sample inefficient.

Feng et al. [40] point out that constant-size logarithmic precision Transformers can implement mathematical and dynamic programming problems. In order to learn a concept with CoT, the data

has to lay out the respective step-by-step solution. Connecting to our bound, the concept needs to be made "obvious" to learn. Various works [17, 41, 42, 43] present approaches to instead allow Transformers to change their depth. This could make the model computationally more powerful as it allows the model to take arbitrary computation time for difficult samples. Our theoretical result instead focuses on fixed-depth Transformers, as they currently are most present in LLMs, and focuses on tasks where the given depth of the Transformers is already sufficient to solve the task.

There has been extensive research on the expressive power of language models [40, 44, 41]. Judd [45] already showed that finding weights for feedforward neural networks such that the neural network correctly predicts at least $^2/_3$ of the training examples is NP-hard in general [46]. Our theoretical bounds work focuses on overparameterized models that memorize. Other follow-ups construct concrete example network architectures being hard to train [46, 47, 48]. Kearns [49] presents results on the complexity of learning, with a focus on learning from distributions, compared to us investigating data and model efficiency. Abbe and Sandon [50] show that decision problems learnable in polynomial time are learnable by gradient descent in polynomial time. Different from us, they give their learning algorithm arbitrarily many samples and distinguish poly-time vs. non-poly-time.

## 7 Conclusions, limitations, and future work

**Limitations.** We do not propose practical ideas to tackle the issues identified with the investigated algorithmic tasks. Hence, it remains for future research to find effective means to incentivize compositional learning in Transformer-based language models, possibly by incorporating more inductive biases towards compositionality in the model architecture or the training procedure. Additionally, while the investigated algorithmic tasks allow to work in a fully controlled environment with limited confounding factors, they still represent a rather limited collection of benchmarks. Its expansion with more natural tasks, as well as the ablation of different sub-task definitions, could strengthen the observations and conclusions proposed in this work.

**Conclusions and future work.** In this work, we analyzed the capabilities of Transformer language models in learning compositional algorithmic tasks. To do so, we formulated a set of hypotheses to characterize the efficiency of a model when learning compositional tasks, from efficient ($\mathcal{H}_1$) to inefficient ($\mathcal{H}_4$) regimes. We introduced a new set of algorithmic tasks based on pointer execution to benchmark compositional learning in large language models in a more controlled setting. On these novel tasks, as well as other well-known benchmarks from previous works, we observe that Transformer-based language models struggle at compositional learning on tasks demanding to learn the composition of several discrete sub-tasks, making our hypothesis $\mathcal{H}_4$ most plausible (learning the compositional task requires more samples than the sum of those required to learn the individual sub-tasks). Further, we show (also theoretically) that learning certain (compositional) concepts with feedforward models on gradient descent is very data inefficient, adding further support to reject $\mathcal{H}_1$–$\mathcal{H}_3$. Finally, we empirically rule out also the possibility that these tasks can be learned in a few-shot fashion by state-of-the-art LLMs such as GPT-4 and Gemini-Pro. This evaluation of current state-of-the-art Transformer language models may provide some directions to improvements in compositional capabilities which would help models to be more reliable and improve on complex algorithmic structures like reasoning or mathematics. Addressing the current limitations of this work might also be an interesting avenue for future research.

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

# Appendix

## Table of Contents

# A    Feedforward models on gradient descent can only learn the obvious[2]

As in our empirical results, we focus on model size and dataset size, not compute, which e.g. is the focus of [51, 52, 50]. This has the rationale, that concepts or tasks which do not have unlimited amounts of data and are hard to learn, can in theory still be learned while gradient descent on feedforward models will be limited. In particular, in this part we show classes of tasks that are learnable from a single sample each (i.e., learnable within hypothesis $\mathcal{H}_1$), but on which feedforward models trained with gradient descent will learn within hypothesis $\mathcal{H}_4$, needing up to exponentially many samples. See also Figure A.4 for an intuition.

From complexity theory, we know that finding the proof of membership of a string in a language is in general less efficient than verifying its membership given a proof. We use this structure:

**Definition A.1** ($\mathcal{K}$)**.** Let $L$ be any recursive decision problem with solution in space $O(n)$ and being verifiable in time $O(n)$. Let us look at all algorithms solving $L$ and let $k(n)$ be a (high) lower bound of them. We denote $\mathcal{K}$ to be the set of choosing one upper bound $k(n)$ per such decision problem.

Some examples: Assuming the exponential time hypothesis, SAT solution algorithms need exponential time (i.e. $k(n)$ can be an arbitrarily high polynomial), and verification is in $O(n)$. Under the same assumption, the edit distance algorithm needs $\Omega(n^{2-\epsilon})$ for any $\epsilon > 0$ while being verifiable in $O(n)$ [53]. Usually, no lower bounds are proven for such problems, but the fact that we can not find faster algorithms makes assumptions reasonable.

**Definition A.2** (Concept, Concept Class)**.** A concept $\mathcal{C} = (U_\mathcal{C}, f_\mathcal{C}, n_\mathcal{C})$ consists of an algorithm $f$ defined on a set of inputs $U$ and a natural number $n_\mathcal{C}$. We call $n_\mathcal{C}$ the "difficulty" of $\mathcal{C}$. A concept class $\mathbb{C}$ is an infinite set of concepts where each $n \in \mathbb{N}$ occurs only finitely many times.

The intuition: A concept class is a set of concepts that are connected by some common basis, e.g. they all pose certain challenges (e.g. finding a formula assignment) to an overarching problem to learn (e.g. the SAT problem). We need a class of such problems because complexity theory makes asymptotic statements.

Similar definitions are used by Kearns [49], here we use a slightly different definition of concepts as functions instead of subsets of an object set to better suit today's applications.

**Definition A.3** (Application of a Concept)**.** Given a concept $\mathcal{C} = (U, f, n)$, an application of $\mathcal{C}$ is a pair $(T, E)$ of nonempty sets of samples (i.e. input-output pairs) of $f$. We call $T$ the training set and $E$ the test set or examination set.

Here we deviate from classical learning theory in that we do not define a distribution. Instead, we define the more general case of having a training regime and testing regime since we focus on systematic generalization. One can additionally require $E$ and $T$ to be statistically dependent to recover classical train-test distributions, as we, for example, show in Appendix A.5.

**Definition A.4** (Learning Algorithm)**.** Let $\mathbb{S}$ denote the set of all finite datasets (i.e., sets of input-output pairs) and $\Theta$ the set of all input-bounded and runtime-bounded algorithms. A learning algorithm is an algorithm $\mathcal{L} : \mathbb{S} \to \Theta$.
We say a $\mathcal{L}$ learns a concept $\mathcal{C} = (U_\mathcal{C}, f_\mathcal{C}, n_\mathcal{C})$ on a dataset $T_\mathcal{C}$, if $\mathcal{M}(x) = f(x) \, \forall x \in U$ with $\mathcal{M} := \mathcal{L}(T_\mathcal{C})$.

Let $\text{TIME}(\mathcal{A}, x)$ denote the runtime of an algorithm $\mathcal{A}$ on an input $x$. For a concept class $\mathbb{C}$ and a learning algorithm $\mathcal{L}$, we denote:

$$z_\mathbb{C}^\mathcal{L}(n_0) = \max_{\substack{(U,f,n_0)\in\mathbb{C} \\ (x,y)\in E_{(U,f,n_0)}}} \text{TIME}(\mathcal{L}(T_{(U,f,n_0)}), x) \tag{2}$$

I.e. the maximum inference runtime on a test sample for concepts of difficulty $n_0$.

**Theorem A.5.** *For each $k \in \mathcal{K}$ there is a (different) concept class $\mathbb{C}$, with applications $(T_\mathcal{C}, E_\mathcal{C})$ (i.e., train and test datasets) for each $\mathcal{C} \in \mathbb{C}$, with the following properties:*

1. *There is a learning algorithm $\mathcal{L}^*$ that can learn each concept from a single sample in $O(k(n_\mathcal{C}))$ time and with $z_\mathbb{C}^{\mathcal{L}^*}(n_\mathcal{C}) = O(n_\mathcal{C})$*

---

[2]Obvious here means: either present in many samples, or simple to extract

2. *For any learning algorithm $\mathcal{L}$, $\mathcal{L}$ one of the following will hold:*

   - *"$\mathcal{L}$ does not learn": $\mathcal{L}$ is not able to learn infinitely many concepts in $\mathbb{C}$.*
   - *"$\mathcal{L}$ learns during inference": With $\mathcal{M} = \mathcal{L}(T_{\mathcal{C}})$, $\sum_{(x,y) \in E_{\mathcal{C}}} \text{TIME}(\mathcal{M}, x) = \Omega(k(n_{\mathcal{C}}))$*
   - *"$\mathcal{L}$ does learn": $\text{TIME}(\mathcal{L}, T_{\mathcal{C}}) = \Omega(k(n_{\mathcal{C}}))$*

The intuitive claim of this theorem is, that there are concepts, which require a $\Omega(k(n_C))$ time (i.e. a lot) to learn while being fast at inference.

A special case for SAT is carried out in Appendix A.3 and the general case in Appendix A.4. The proof focuses on training on a minimal dataset (one sample) and testing on an out-of-distribution "exam" set. For completeness, in Appendix A.5 we provide a proof for similar claims in an in-distribution training-testing setting.

We now introduce a memorization assumption which assumes that our learning algorithm memorizes fast.

**Definition A.6** (Constant Memorization). Given a concept class $\mathbb{C}$ with applications $T_{\mathcal{C}}, E_{\mathcal{C}}$ for each $\mathcal{C} \in \mathbb{C}$ and a learning algorithm $\mathcal{L}$. We say $\mathcal{L}$ memorizes constantly under $\mathbb{C}$, if there is a constant $B$, such that for any $T_{\mathcal{C}}, E_{\mathcal{C}}$, $\mathcal{L}$ runs in time $B * |T_{\mathcal{C}}| * Z$ with $Z$ being the longest runtime of $\mathcal{M} := \mathcal{L}(T_{\mathcal{C}})$ on a sample in $E$.

With this, we obtain:

**Corollary A.7.** *Let $k \in \mathcal{K}$. Then there exists a concept class $\mathbb{C}$ with applications $T_{\mathcal{C}}, E_{\mathcal{C}}$ such that:*

1. *Any constantly memorizing learning algorithm $\mathcal{L}$ learning on an (arbitrary) augmented dataset $T'_{\mathcal{C}}$ generated by a generator $T'_{\mathcal{C}} = G(T_{\mathcal{C}})$ with $G$ running in time $o(k(n_{\mathcal{C}}))$: If $z^{\mathcal{L}}_{\mathbb{C}}(n_{\mathcal{C}}) = o(\frac{k(n_{\mathcal{C}})}{n_{\mathcal{C}}})$, then $\mathcal{L}$ needs $\Omega(\frac{k(n_{\mathcal{C}})}{z^{\mathcal{L}}_{\mathbb{C}}(n_{\mathcal{C}})})$ many samples to learn each concept $\mathcal{C} = (U_{\mathcal{C}}, f_{\mathcal{C}}, n_{\mathcal{C}}) \in \mathbb{C}$.*

2. *There exists a learning algorithm $\mathcal{L}^*$ learning each concept from a single sample with $z^{\mathcal{L}^*}_{\mathbb{C}}(n_{\mathcal{C}}) = O(n_{\mathcal{C}})$.*

We use the notion of $G$ (which is an algorithm free to choose) to make sure that bigger datasets do not make things "obvious", that is, leak too much information about the exam $E_{\mathcal{C}}$.

If we think about the PEN task, it is a combination of sub-tasks, i.e., $PEN(x) = Cpy(PE(RCpy(x)))$. There are combinations for three of the sub-tasks chained together. Already in such a restricted setting, learning the task from scratch is extremely inefficient, requiring a multiplicative factor of one million samples more than the discrete optimization algorithm described in Appendix B.2, which in this case corresponds to the optimal learner mentioned in Corollary 5.7. This confirms the outcome that our theoretical framework would predict, i.e. LLaMA fails at learning from few examples, although the existence of the optimal learner shows that it would be possible.

The theory allows us to extend this result beyond the limit of the empirically verifiable. When more primitives are available (e.g. in general-purpose datasets like The Pile [54], finding the right functional composition of the sub-tasks quickly becomes a computationally hard combinatorial task. With such problems, the theory says that plain gradient descent will tend to just memorize the few samples of the compositional task, instead of (combinatorially) finding the right composition.

Nasr et al. [55] show that one can extract more memorized data from larger models. Tirumala et al. [56] have shown that larger language models memorize faster than smaller models. Based on this, we can conservatively assume a constant number of steps for memorizing the fixed-size answer to a sample. This fits definition A.6 for a learning algorithm $\mathcal{L}^F$ learning with (growing) constant-depth feedforward models, depending on the input size in the training data, as one gradient step has the time complexity of a forward pass.

Therefore, Corollary A.7 can be interpreted as follows: There are classes of concepts, such that to learn them reliably, either the feedforward deep learning model needs to be unfavorably larger than needed for inference ($z^{\mathcal{L}^F}_{\mathbb{C}}(n) >> n$), or the data inefficiency is very high (by a factor $\frac{k(n)}{z^{\mathcal{L}^F}_{\mathbb{C}}(n)}$). Note that in practice, memorization does not necessarily mean, we stop optimizing, here we implicitly assume that after memorization, nothing useful happens anymore, e.g., the gradient signal magnitude is 0 and the model does not change further.

Much research has been done on the complexity of various problems. Here we want to point out a few further concept classes that need significant computation to learn and therefore underlie the above statements:

A fundamental capability for safe, explainable, and intelligent systems, is to be able to infer causal relationships. It has been shown, however, that learning such structural causal models is NP-hard in the number of graph nodes [57, 58]. Therefore: Let our complexity class contain the functions applying all structural causal models and generate observations for each concept. Now let us test those functions by an exam after training (i.e. by just asking for the structural causal model). Then, for this specific concept class, Theorem A.5 and Corollary A.7 hold for $k(n)$ being a super-polynomial function.

Further, it has been shown [59] that given a structural causal model, it is NP-hard to infer whether one particular event always leads to another. Such inference might be important for a few selected cases. Again, by constructing a concept class containing functions such that the structural model is given or inferable and the always-cause is asked or needed for predictions in the inference phase, it follows by Theorem A.5 and Corollary A.7 that learning such concepts requires a lot of computation - and many samples for feedforward neural networks although theoretically unnecessary.

## A.1 Details on the theory

For brevity "algorithm" refers to always terminating algorithms. Let us fix a standard random access memory computational model. We also fix a universal encoding and decoding function ENC, DEC that can encode/decode any algorithm or object we define below (imagine, for example, a digital processor model where everything can be encoded in 0s and 1s). From now on we omit all encodings and decodings and instead use defined objects and their encodings interchangeably for the sake of simple notation.

## A.2 Intuition: Learning-inference gap

See Figure A.4.

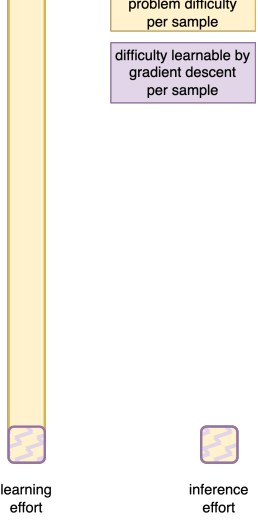

Figure A.4: Intuition for our statement: Based on widely accepted assumptions, there are problems requiring much more learning computation time than usage computation time (here: per sample available). Gradient Descent on Feedforward Networks under the assumption of constant-step-memorization however, will only be able to learn problems as hard to learn as they are during inference. Therefore, the model size needs to be much larger than necessary, or the training samples need to be much larger than necessary, or gradient descent will have trouble learning.

## A.3 Special case: SAT proof on algorithmic learning

For a better reader's experience, let us restate the theorem:

**Theorem.** *For each $k \in \mathcal{K}$ there is a (different) concept class $\mathbb{C}$, with applications $(T_{\mathcal{C}}, E_{\mathcal{C}})$ (i.e., train and test datasets) for each $\mathcal{C} \in \mathbb{C}$, with the following properties:*

*1. There is a learning algorithm $\mathcal{L}^*$ that can learn each concept from a single sample in $O(k(n_{\mathcal{C}}))$ time and with $z_{\mathbb{C}}^{\mathcal{L}^*}(n_{\mathcal{C}}) = O(n_{\mathcal{C}})$*

*2. For any learning algorithm $\mathcal{L}$, $\mathcal{L}$ one of the following will hold:*

- *"$\mathcal{L}$ does not learn": $\mathcal{L}$ is not able to learn infinitely many concepts in $\mathbb{C}$.*
- *"$\mathcal{L}$ learns during inference": With $\mathcal{M} = \mathcal{L}(T_{\mathcal{C}})$, $\sum_{(x,y) \in E_{\mathcal{C}}} \text{TIME}(\mathcal{M}, x) = \Omega(k(n_{\mathcal{C}}))$*
- *"$\mathcal{L}$ does learn": $\text{TIME}(\mathcal{L}, T_{\mathcal{C}}) = \Omega(k(n_{\mathcal{C}}))$*

For simplicity, we first prove a special case where $k$ is the minimal runtime of a SAT (3-satisfiability) problem (i.e. given $P \neq NP$ exponential in $n$).

*Proof.* Valiant and Vazirani [60] have shown, that given a formula which is either uniquely satisfiable or not satisfiable, deciding this is NP-hard. From this follows, that given a uniquely satisfiable formula, it is (assuming $P \neq NP$) not possible in polynomial time, to find a proof for it, i.e. the unique satisfying variable assignment (this holds, because checking such a proof is fast).

We choose $\mathbb{C}$ to be a set constructed from the set of unambiguously satisfiable SAT formulas: Given a uniquely satisfiable formula $A$ and the unique solution $y^A$, let $U^A = \{A, 1, \ldots, |y^A|\}$ be the set containing the formula and the variable names. Let $f^A(A) = 1$ and $f^A(i) = y_i^A$. Our concept is now $\mathcal{C}^A = (U^A, f^A, |A|)$. We choose the training dataset of the concepts to be $\{(A, 1)\}$ and the examination dataset to be $\{(i, y_i^A) | i \in 1, \ldots, |y^A|\}$. This means we train on a uniquely satisfiable formula, and the test set tests whether the model learned the unique solution to it.

We now construct the learning algorithm $\mathcal{L}^*$ learning $\mathcal{C} \in \mathbb{C}$ from one sample. Let us fix an arbitrary $(U_{\mathcal{C}}, f_{\mathcal{C}}, n_{\mathcal{C}}) \in \mathbb{C}$.

Given the training dataset $T_{\mathcal{C}} = \{(A_{\mathcal{C}}, 1)\}$ corresponding to $\mathcal{C}$, $\mathcal{L}^*$ builds an algorithm $\mathcal{M}$ of the form:

- $\mathcal{M}(A) = \text{CHECK}(A, x)$ for a formula $A$ of length $n_{\mathcal{C}}$ where $x, |x| \leq n_{\mathcal{C}}$ is a stored (i.e. hardcoded) constant. CHECK returns 1 iff $x$ is a satisfying variable assignment for the variables of $A$. Note that $n_{\mathcal{C}}$ can be inferred by $\mathcal{L}^*$ as it is the length of the single sample in $T_{\mathcal{C}}$.

- $\mathcal{M}(i) = x_i$, i.e. the $i$-th variable assignment.

- On all other inputs, $\mathcal{M}$ can implement any algorithm (e.g. some pre-trained knowledge).

Note that $\mathcal{M}$ has a linear running time in $n_{\mathcal{C}}$ on all samples in $T_{\mathcal{C}}$ and $E_{\mathcal{C}}$. Now, $\mathcal{L}^*$ finds the satisfying variable assignment $x$ (using the minimal-runtime algorithm) for $A_{\mathcal{C}}$ and stores it in $x$. Note that this procedure runs in $k_{SAT}(n_{\mathcal{C}})$ for $k_{SAT}(n_{\mathcal{C}})$ being the minimal runtime for finding a solution to uniquely satisfiable formulas (which is assumed to be exponential under the exponential time hypothesis).

Now to the second step: Assume there is a learning algorithm $\mathcal{L}$ and assume that it can learn all concepts except finitely many and do so in $o(k_{SAT}(n_{\mathcal{C}}))$ (i.e. strictly less) time and with $\mathcal{M} = \mathcal{L}(T_{\mathcal{C}})$ needing $o(k_{SAT}(n_{\mathcal{C}})/n_{\mathcal{C}})$ time on each sample in $E_{\mathcal{C}}$. Then one can simulate the learning setting to find a solution to a formula $A$, $n := |A|$ (except finitely many times): First, simulate $\mathcal{L}$ on $\{(A, 1)\}$ to obtain $\mathcal{M} = \mathcal{L}(\{(A, 1)\})$. Then, simulate $\mathcal{M}$ on all variable indices $i$ as inputs to obtain the satisfying assignment $y^A$. This algorithm finds $y^A$ in $o(k_{SAT}(n)) + o(n * \frac{k_{SAT}(n)}{n}) = o(k_{SAT}(n))$ time, contradiction.

$\square$

## A.4 General proof on algorithmic learning

This proof is analogous to the special SAT case. We recommend the reader to read the special case first.

*Proof.* Let $k \in \mathcal{K}$. Let $\mathcal{P}$ be the problem $k$ corresponds to (e.g. SAT). Let $\mathcal{A}_s$ denote the solving algorithm for $\mathcal{P}$ running in time $k(n)$, i.e. an algorithm that computes a solution to each instance of a class of problem cases, where the solution is in $O(n)$ space. Let $\mathcal{A}_c$ be the checking algorithm that can verify if such a solution is correct, running in $O(n)$. We choose the concept class $\mathbb{C}$ to be a set constructed from the set of positive instances in $\mathcal{P}$ (e.g. in the SAT special case this would be a uniquely satisfiable CNF formula): Given such a positive input instance $X$, let $U^X = \{X, 1, \ldots, |\mathcal{A}_s(X)|\}$. Let $f^X(X) = 1$ and $f^X(i) = \mathcal{A}_s(X)_i$. Our concept is now $\mathcal{C}^X = (U^X, f^X, |X|)$. We choose the training dataset of the concepts to be $\{(X, 1)\}$ and the examination dataset to be $\{(i, y_i^X | i \in 1, \ldots, |y^X|\}$.

We now construct the learning algorithm $\mathcal{L}^*$ learning $\mathcal{C} \in \mathbb{C}$ from one sample. Let us fix an arbitrary $(U_\mathcal{C}, f_\mathcal{C}, n_\mathcal{C}) \in \mathbb{C}$.

Given the training dataset $T_\mathcal{C} = \{(X, 1)\}$ corresponding to $\mathcal{C}$, $\mathcal{L}^*$ builds an algorithm $\mathcal{M}$ of the form:

- $\mathcal{M}(X) = \text{CHECK}(X, s)$ for a problem instance $X$ of the problem $\mathcal{P}$ with $|X| \leq n_C$. $|s| = O(n_\mathcal{C})$ is a stored (i.e. hardcoded) constant. CHECK returns 1 iff $s$ is a proof for the problem instance $X$ (which exists by Definition A.1). Values for $s$ will be candidate solutions for $X$, and $\mathcal{L}^*$ will search for the solution.

- $\mathcal{M}(i) = s_i$, i.e. the $i$-th value (e.g. bit) of $s$.

- On all other inputs, $\mathcal{M}$ can implement any algorithm (e.g. some pre-trained knowledge).

Note that $\mathcal{M}$ has a linear runtime in $n_\mathcal{C}$ on all samples in $T_\mathcal{C}$ and $E_\mathcal{C}$. Now, $\mathcal{L}^*$ computes $s^* = \mathcal{A}_s(X)$ (i.e. the solution to $\mathcal{P}$) and stores it in $s$, i.e. $s = s^*$. Note that this procedure runs in $\text{TIME}(A_s, X)$ time (which can be $k(n_C)$ if $k(n_C)$ is tight).

Now to the second step: Assume there is a learning algorithm $\mathcal{L}$ and assume that it can learn all concepts except finitely many and do so in $o(k(n_\mathcal{C}))$ (i.e. strictly less) time and with $\mathcal{M} = \mathcal{L}(T_\mathcal{C})$ needing $o(k(n_\mathcal{C})/n_\mathcal{C})$ time on each sample in $E_\mathcal{C}$. Then one can simulate the learning setting to find a solution to a problem case $X \in \mathcal{P}$, $n := |X|$ (except finitely many times): First, simulate $\mathcal{L}$ on $\{(X, 1)\}$ to obtain $M = \mathcal{L}(\{(X, 1)\})$. Then, simulate $\mathcal{M}$ on all solution indices as inputs to get the proof $y^X$ for $X$. This algorithm finds $y^X$ in $o(k(n)) + o(n * \frac{k(n)}{n}) = o(k(n))$ time, contradiction.

$\square$

## A.5 Algorithmic learning in the in-distribution setting

The underlying idea of this proof is to use one-way functions that are assumed to be hard even when one sees many examples. Such functions are known from cryptography. By using public-key cryptography we enable an attacker to simulate our learning setting which then allows us to show that this learning setting cannot be successful easily while we make sure, the setting is principally learnable [49]. For more elaborate motivations, we refer to this work.

Assumption $A.8$ is a standard assumption about cryptography being safe.

**Assumption A.8** (Randomized RSA is CPA Secure)**.** Let $pk, pk^{-1}$ denote an RSA encryption scheme with padding the plaintext with zeros and random bits before encryption. Assume that this encryption algorithm is CPA secure.

**Theorem A.9.** *Let $k(n)$ be any polynomial function. Assuming Assumption A.8, there exists a randomized concept class $\mathbb{C}$ containing exactly one concept $\mathcal{C}_n$ per $n \in \mathbb{N}^+$, with applications $(T_\mathcal{C}, E_\mathcal{C})$ (i.e. train and test datasets) where $T_\mathcal{C}$ and $E_\mathcal{C}$ are drawn from the same distribution, with the following properties:*

- *There is a learning algorithm that can learn each concept from a single sample.*

- *For any learning algorithm $\mathcal{L}$ and any $n$, $\mathcal{L}$ will either have negligible (in $n$) probability of learning concept $\mathcal{C}_n$. Otherwise, $\mathcal{M} = \mathcal{L}(T_{\mathcal{C}_n})$ needs $\Omega(\frac{k(n)}{n})$ computation time on a sample in $E_{\mathcal{C}_n}$, or $\mathcal{L}$ needs $\Omega(k(n))$ computation time to learn.*

*Proof.* Let us denote it by $pk, pk^{-1}$ our randomized RSA encryption scheme. For each $n \in \mathbb{N}$, we now build one concept:

First let $s \in \{0,1\}^n$ be the randomly chosen secret for $pk, pk^{-1}$. We define:

$$U_+ = \{(x, pk(x), pk) | x \in \{0,1\}^n\} \tag{3}$$
$$U_- = \{(y, pk(x), pk) | y \in \{0,1\}^m, x \in \{0,1\}^n\} \setminus U_+ \tag{4}$$

where $m$ denotes the ciphertext size of $pk$. Note that the encryption is randomized and $U_+$ contains all combinations $pk(x)$ for the random bits. We define $f_n(e) = 1$ if $e \in U_+$ and $f_n(e) = 0$ if $e \in U_-$. The concept is now $(U_+ \cup U_-, f_n, n)$. Let us fix some constant size for $T_n, E_n$ each (one is sufficient). Each sample in $T_n$ and $E_n$ is drawn with probability $\frac{1}{2}$ from $U_+$ u.a.r. and with probability $\frac{1}{2}$ from $U_-$ u.a.r.

With $\mathbb{C}$ being defined, let us construct the learning algorithm $\mathcal{L}^*$ learning from one sample. Let $n$ be fixed and sample $T_n, E_n$. $\mathcal{L}^*$ chooses an arbitrary sample $(x, pk(x), pk)$ and brute-forces the secret key $d$ of the RSA encryption scheme. Note that this key is unique given the public key information $pk$. $\mathcal{L}^*$ now builds an algorithm $\mathcal{M}$ for the form:

- $\mathcal{M}(x, c, pk) = 1$ iff the decoding of $c$ is $x$ and 0 otherwise

- On all other inputs, $\mathcal{M}$ can implement any algorithm (e.g. some pre-trained knowledge).

Note that those operations are efficiently doable, much faster than breaking the RSA secret key.

Now to the second part: Let $k$ be any polynomial function and fix some $n$. Assume there is a learning algorithm $\mathcal{L}$ and assume that it learns the concept $C_n$ with non-negligible probability in $n$. Further assume that $\mathcal{L}$ needs $o(k(n))$ time and $\mathcal{M} = \mathcal{L}(T_{\mathcal{C}_n})$ spends $o(\frac{k(n)}{n})$ for each sample $s \in E_\mathcal{C}$. Then, we can break RSA encryption in polynomial time: Let our encryption-breaking algorithm $\mathcal{E}$ be given a public key configuration (including $n$). It samples a training dataset $T$ with the public key. Then it simulates $\mathcal{L}$ on it and obtains $\mathcal{M} = \mathcal{L}(T)$. For the IND-CPA game, it chooses words $m_0 \neq m_1$ randomly and gets the (randomized) encryption $c$ of one of them back. $\mathcal{E}$ sets $\{(m_0, c, pk)\}$ as the test set. $\mathcal{E}$ now computes the answer by simulating the trained model and choosing $m_0$ if $\mathcal{M}((m_0, c, pk)) = 1$. This will give $\mathcal{E}$ non-negligible success probability for the IND-CPA game in polynomial time, contradiction.

$\square$

## A.6 Proof of the corollary A.7

**Corollary.** *Let $k \in \mathcal{K}$. Then there exists a concept class $\mathbb{C}$ with applications $T_\mathcal{C}, E_\mathcal{C}$ such that:*

- *Any constantly memorizing learning algorithm $\mathcal{L}$ learning on an (arbitrary) augmented dataset $T'_\mathcal{C}$ generated by a generator $T'_\mathcal{C} = G(T_\mathcal{C})$ with $G$ running in time $o(k(n_\mathcal{C}))$: If $z_\mathbb{C}^\mathcal{L}(n_\mathcal{C}) = o(\frac{k(n_\mathcal{C})}{n_\mathcal{C}})$, then $\mathcal{L}$ needs $\Omega(\frac{k(n_\mathcal{C})}{z_\mathbb{C}^\mathcal{L}(n_\mathcal{C})})$ many samples to learn each concept $\mathcal{C} = (U_\mathcal{C}, f_\mathcal{C}, n_\mathcal{C}) \in \mathbb{C}$.*

- *There exists a learning algorithm $\mathcal{L}^*$ learning each concept from a single sample with $z_\mathbb{C}^{\mathcal{L}^*}(n_\mathcal{C}) = O(n_\mathcal{C})$.*

*Proof.* Let us fix $k \in \mathcal{K}$ and define our concept class $\mathbb{C}$ with applications $E_\mathcal{C}, T_\mathcal{C}$ as given by Theorem A.5.

The second part follows immediately by Theorem A.5.

The first part: Let $\mathcal{L}$ be a constantly memorizing learning algorithm $\mathcal{L}$ with a generator $G$. Assume that $z_\mathbb{C}^\mathcal{L}(n_\mathcal{C}) = o(\frac{k(n_\mathcal{C})}{n_\mathcal{C}})$ and, $\mathcal{L}$ learns with $|T'_\mathcal{C}| = o(\frac{k(n_\mathcal{C})}{z_\mathbb{C}^\mathcal{L}(n_\mathcal{C})})$. Then by Definition A.6, there exists a constant $B$ such that for every $\mathcal{C} \in \mathbb{C}$, $\mathcal{L}$ runs in time:

$$B * |T'_\mathcal{C}| * Z_\mathcal{C} \leq B * |T'_\mathcal{C}| * z_\mathbb{C}^\mathcal{L}(n_\mathcal{C}) = o(\frac{k(n_\mathcal{C})}{z_\mathbb{C}^\mathcal{L}(n_\mathcal{C})} * z_\mathbb{C}^\mathcal{L}(n_\mathcal{C})) = o(k(n_\mathcal{C})).$$

Therefore, we can summarize that $G$ runs in $o(k(n_\mathcal{C}))$, $\mathcal{L}$ runs in $o(k(n_\mathcal{C}))$, and $\mathcal{M} = \mathcal{L}(T'_\mathcal{C})$ runs in $z_\mathbb{C}^\mathcal{L}(n_\mathcal{C}) = o(\frac{k(n_\mathcal{C})}{n_\mathcal{C}})$. This can't be, as Theorem A.5 states that we need more computation time. $\qquad\square$

# B    Specifications of PEN, PERM, HSS, and MUL

## B.1    Properties of the sub-task definition

In this Appendix section, we expose more details about our definition of the set of sub-tasks $\mathcal{S}$ and the properties that it shows. In particular, we characterize it through different angles, namely:

- **atomicity**: each sub-task is an atomic unit roughly corresponding to a basic block of operations in the equivalent procedural programming language.

- **minimality**: among all the possible definitions of sub-tasks, we define $\mathcal{S}$ to be the minimal set of sub-tasks, where every primitive operation is made observable in *exactly* one sub-task.

- **usefulness**: the defined set of primitives is useful to learn the compositional task. We validate this in our experiments on task de-composition (Section 4.2), which shows that the model learns and compose the given set of primitives when it is trained on the full compositional task.

- **uniqueness**: by imposing independent observability on the different primitives, we can relax the constraint on the bijection between primitives and sub-tasks, allowing for sub-tasks that contain more than a single primitive. As a result, the definition of $\mathcal{S}$ is not unique. However, this does not undermine the validity of the investigation, since we are measuring a *relative* phenomenon. The inefficiency of Transformer-based language models that we observe is always relative to the considered set of chosen sub-tasks. However, considering all the properties above, we speculate that similar results could be also measured for alternative definitions of the sub-tasks set $\mathcal{S}$.

## B.2    Details on the PEN task

In addition to the task description in Section 3, there are some additional constraints, visualized in Figure B.5:

1. The (yellow) neighbors of the green chain build a chain themselves, starting at the second word in the sequence (i.e. the first yellow word).

2. The full sequence is at least double the length of the green matching sequence. With all other 'free' green words, we build another matching sequence starting at a random "free" green position (we call it the "free" matching sequence, see the blue arrows in Figure B.5). This is important because then the model cannot easily tell apart the green positions of the correct matching sequence with the green positions of the "free" matching sequence without either matching all green tokens from the start or looking at the answer words and starting matching from their left neighbors.

3. Each yellow word next to a true green matching word (except the terminating word in the yellow matching sequence) has a doppelganger (which we also call attention trap) next to a "free" green word. We add this to "confuse" the model: since the model learns to match words to words, it could be tempted to match yellow words, which would give it the wrong order of neighbors. To make them distinguishable in the answer, we define the doppelganger to match equally to the previous and next word but have a different "data" number in the middle.

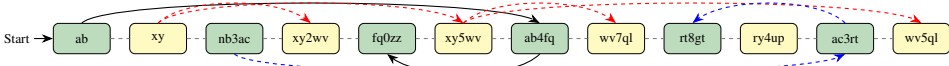

Figure B.5: Matchings of the yellow words: The (yellow) neighbors of matched green words build a matching sequence themselves (in their own order) and have exactly one matching outside the neighbors each (except the last yellow word in the order of the yellow sequence, which has no matching). Therefore, each yellow neighbor except the last matches to two words in the sequence. The blue arrows are over the remaining green positions (which are not part of any answer), which also build a matching sequence. Those additional constraints remove shortcut solutions.

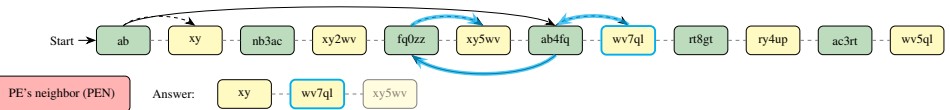

Figure B.6: How a language model with next-token-prediction can solve the PEN task. Take the most recent word in the answer, find its left neighbor ("left"), find the match ("match"), and retrieve its right neighbor ("right"). This corresponds to the sub-tasks RCpy (e.g., in this task, the model has to find left neighbors), PE, and Cpy. The data efficiency in de-compositionality (see Tables C.10, C.11) provide some evidence that this is the natural way a model solves the task when being close to $100\%$ performance and without being trained on the sub-tasks.

# C  Training experiments details

## C.1  LLaMA training hyperparameters

We use a batch size of $384$ samples, which corresponds to ca. $250\,K$ tokens per batch for the PEN task and $50\,K$ for the PERM task. Our $150\,M$ parameter model contains of 12 layers and a hidden size of $1024$. The learning rate is $10^{-4}$, as in the original paper [2].

## C.2  Details on the discrete solver of the PEN Task

Our discrete learning algorithm which can learn the PEN task from a single example is a 2-gram model over 11 words resulting in 121 parameters with $8$ values each. This results in a search space of size $1.9 * 10^{109}$. Each value is an action function that manipulates the current word in the forward pass of the model (at the beginning of the forward pass it is set to "error") and gives back one of the 11 action outcomes. The last two action outcomes (initially (BOS,BOS)) result in the next 2-tuple determining the next discrete parameter. As this execution could loop or run over many parameters, we limit the depth to 12. The actions and their outcomes are:

- EOS: retrieves the eos constant (outcome: EOS)
- LAST-OUTPUT: retrieves the last output word (or the start word if it is the beginning of the answer) (outcome: LAST-OUTPUT)
- MATCH: matches, if there is no next match, this will return error (outcome: MATCH)
- LEFT: get the left neighbor of the current word (outcome: LEFT)
- RIGHT: get the right neighbor of the current word (outcome: RIGHT)
- IS-START: is it the beginning of the answer (outcomes: IS-START-TRUE, IS-START-FALSE)
- IS-ERROR: is the current state error (outcomes: IS-ERROR-TRUE,IS-ERROR-FALSE)
- OUTPUT: output the current word (outcome: OUTPUT)
- For an entry point we define an additional action outcome: BOS

The search algorithm uses a simple hill-climbing heuristic with restarts towards getting as many answer words in the sample right as possible. To search more effectively, we add a penalty to changing the same parameter over and over. The most challenging part of our novel pointer execution tasks is to learn the `left-match-right` structure, which is easily found by this algorithm.

## C.3 Accuracy tables

The numbers are visualized in Section 4. Each row in the tables C.2, C.3 C.4, C.5, C.6, and C.8 corresponds to one run with the respective data mix.

| Model | Cpy | RCpy | PE | PEV | PEN | Cpy | RCpy | PE | PEV | **PEN** |
|---|---|---|---|---|---|---|---|---|---|---|
| | Number of samples (in thousand) | | | | | Accuracy at convergence | | | | |
| LLaMA 150 M | 20 | 20 | 60 | 100 | 100 | 1.00 | 1.00 | 0.99 | 0.91 | **0.00** |
| | 20 | 20 | 60 | 100 | 500 | 1.00 | 1.00 | 1.00 | 0.99 | **0.93** |
| | 20 | 20 | 60 | 100 | 750 | 1.00 | 1.00 | 1.00 | 0.98 | **0.95** |
| | 20 | 20 | 60 | 100 | 1000 | 1.00 | 1.00 | 0.99 | 0.99 | **0.97** |

Table C.2: Performance of LLaMA on in-distribution test samples of the Pointer Execution's neighbor (PEN) task and its sub-tasks. Training is until convergence. We used the standard language modeling loss on every next token in the input, which performed best. The best validation performance is taken for each task separately. As shown, while models learn all sub-tasks Cpy, RCpy, and PE very well, on the PEN task (which is a composition of the sub-tasks Cpy, RCpy, and PE) they need much larger amounts of data than on any of the sub-tasks. This observation makes hypothesis $\mathcal{H}_4$ most plausible for PEN.

| Model | PEN
# samples
(in thousand) | PEN
Accuracy at
convergence |
|---|---|---|
| LLaMA 150 M | 300 | **0.00** |
| | 500 | **0.74** |
| | 750 | **0.98** |
| | 1000 | **0.98** |
| LLaMA 28 M auxiliary loss | 300 | **0.98** |

Table C.3: Results of LLaMA models trained only on the Pointer Execution's neighbor (PEN) task. Compared to Table C.2, we observe that the model does not seem to systematically profit from the compositionality given there.

| Model | PE | PEM | PER | PERM | PE | PEM | PER | PERM |
|---|---|---|---|---|---|---|---|---|
| | Number of samples (in thousand) | | | | Accuracy at convergence | | | |
| LLaMA 150 M | 40 | 40 | 150 | 150 | 1.00 | 1.00 | 0.99 | **0.25** |
| | 40 | 40 | 150 | 230 | 1.00 | 1.00 | 0.99 | **0.30** |
| | 40 | 40 | 150 | 1000 | 1.00 | 1.00 | 1.00 | **0.74** |
| | 40 | 40 | 150 | 1500 | 1.00 | 1.00 | 1.00 | **0.91** |
| | 40 | 40 | 150 | 2000 | 1.00 | 1.00 | 1.00 | **0.91** |

Table C.4: Performance of LLaMA models on in-distribution test samples of the Pointer Execution Reverse Multicount (PERM) task and its sub-tasks. Similar to Table C.2, we observe that although all sub-tasks are learned perfectly by the models, their composition does not learn fully until including an almost 10-fold increase of the data needed for all sub-tasks together, making hypothesis $\mathcal{H}_4$ most plausible for this task too.

| Model | PERM
# samples
(in thousand) | PERM
Accuracy at
convergence |
|---|---|---|
| LLaMA
150 M | 500
1000
1500
2000 | **0.21**
**0.68**
**0.65**
**0.98** |

Table C.5: Results of LLaMA models trained only on the PERM task. Compared to Table C.4, we observe that the model does not seem to systematically profit from the compositionality given there, similar to PEN.

| Model | SSE | HSS | SSE | HSS |
|---|---|---|---|---|
| | samples (in thousand) | | Accuracy at convergence | |
| LLaMA
150 M | 25
25
25
25 | 25
50
100
200 | 1.00
0.99
0.99
0.99 | **0.68**
**0.85**
**0.95**
**0.99** |

Table C.6: Performance of LLaMA models on Highest Subsequence Sum and Highest Subsequence Execution. Hypothesis $\mathcal{H}_4$ is the most plausible.

| Model | HSS
# samples
(in thousand) | HSS
Accuracy at
convergence |
|---|---|---|
| LLaMA
150 M | 25
50
100
200 | **0.67**
**0.84**
**0.95**
**0.99** |

Table C.7: Performance of LLaMA models only on the Highest Subsequence Sum task.

| Model | DMUL | ADD | MUL | DMUL | ADD | MUL |
|---|---|---|---|---|---|---|
| | samples (in thousand) | | | Accuracy at convergence | | |
| | 10 | 190 | 200 | 1.00 | 0.99 | **0.65** |
| LLaMA | 10 | 190 | 400 | 1.00 | 0.99 | **0.68** |
| 150 M | 10 | 190 | 800 | 1.00 | 0.99 | **0.74** |
| | 10 | 190 | 1600 | 1.00 | 0.99 | **0.77** |

Table C.8: Performance of LLaMA models on Digit Multiplication (DMUL), addition (ADD), and multiplication (MUL). Hypothesis $\mathcal{H}_4$ is the most plausible.

| Model | MUL # samples (in thousand) | MUL Accuracy at convergence |
|---|---|---|
| | 200 | **0.56** |
| LLaMA | 400 | **0.63** |
| 150 M | 800 | **0.78** |
| | 1600 | **0.77** |

Table C.9: Performance of LLaMA models only on multiplication (MUL).

## C.4 Performance of LLaMA on decompositionality

In Tables C.10 and C.11 we include more details results about experiments on decompositionality with LLaMa.

| | PEN | | PEV | |
|---|---|---|---|---|
| Model | Pretraining samples (in thousand) | Pretraining Accuracy | Finetuning samples (in thousand) | Finetuning Accuracy |
| LLaMA 150 M | 1000 | 0.98 | 2 | **0.48** |
| | 1000 | 0.98 | 8 | **0.92** |
| | 1000 | 0.98 | 12 | **0.93** |
| | 1000 | 0.98 | 25 | **0.97** |
| | 1000 | 0.98 | 50 | **0.99** |

Table C.10: Performance of LLaMA models pre-trained on Pointer Execution's neighbor (PEN), and after finetuning on Pointer Execution Verbose (PEV) until convergence (i.e., PEN→PEV). We observe that decompositionality significantly improves data efficiency compared to training a model on the finetuning task from scratch. At the same time, however, for learning the task to full performance, it still seems to need more than a low constant number of demonstrations ($\geq 50$ K), making the $\mathcal{H}_1$-equivalent (i.e. "A Transformer language model learns a decomposition with a constant number of samples") rather implausible for decompositionality as well.

| | PERM | | PE | |
|---|---|---|---|---|
| Model | Pretraining samples (in thousand) | Pretraining Accuracy | Finetuning samples (in thousand) | Finetuning Accuracy |
| LLaMA 150 M | 2000 | 0.98 | 1 | **0.80** |
| | 2000 | 0.98 | 3 | **0.96** |
| | 2000 | 0.98 | 5 | **0.97** |
| | 2000 | 0.98 | 10 | **0.98** |
| | 2000 | 0.98 | 20 | **0.99** |

Table C.11: Performance of LLaMA models pre-trained on Pointer Execution Reverse Multicount (PERM), and after finetuning on Pointer Execution (PE) until convergence (i.e., PERM→PE). As shown similarly in Table C.10, decompositionality significantly improves data efficiency compared to training a model on the finetuning task from scratch. For learning the task to full performance, however, it still seems to need more than a low constant number of demonstrations (5–20 K), making the $\mathcal{H}_1$-equivalent "A Transformer language model learns a decomposition with a constant number of samples" rather implausible for decompositionality.

## C.5 Model ablation: Performance of UT-style LLaMA on PERM

We ablate a variation of our model architecture based on the Universal Transformer (UT) Csordás et al. [61] and the more recent Hyper-UT [17], for both of which there is evidence of better compositional generalization than the original Transformer. For instance, a UT-style transformer displayed good performance on compositional tasks like SCAN Csordás et al. [61]. The model based on UT reached a similar performance as the original LLaMA model, i.e. a behavior that could be classified in $\mathcal{H}_4$. We then also experimented with a Hyper-UT-style LLaMA, based on the aforementioned Hyper-UT work [17], in which each layer selects its weights from a common weight embedding pool to realize a parameter-efficient model. This model consistently achieved a lower accuracy compared to LLaMA on a large set of small algorithmic tasks. Thus, we chose not to systematically explore its performance in a more challenging compositional algorithmic setting. The results are shown in table C.12. The hyperparameters used to train the model are the same as the ones used for the standard models (see Appendix C.1).

| Model | PE | PE Multi | PER | PERM | PE | PE Multi | PER | PERM |
|---|---|---|---|---|---|---|---|---|
| | Number of samples (in thousand) | | | | Accuracy at convergence | | | |
| | 40 | 40 | 150 | 150 | 1.00 | 0.99 | 0.93 | **0.26** |
| LLaMA | 40 | 40 | 150 | 230 | 1.00 | 0.99 | 0.98 | **0.29** |
| 150 M | 40 | 40 | 150 | 1500 | 1.00 | 1.00 | 1.00 | **0.98** |
| | 40 | 40 | 150 | 2000 | 1.00 | 1.00 | 1.00 | **0.99** |

Table C.12: Performance of LLaMA with weight sharing across all transformer layers on the PERM task and its sub-tasks. We see similar results compared to Table C.4, i.e., hypothesis $\mathcal{H}_4$ is most plausible for the PERM task on this model.

# D  Prompting experiments details

## D.1  Input sequences tokenization

Unlike open-source models, such as the LLaMa, closed-source LLMs do not allow to customize the tokenization process. Hence, tokenization could potentially represent a major confounder for our prompting experiments. If fact, an erroneous mapping between words and tokens would inevitably undermine the ability of the model to correctly perform matching operations in the input sequence, making it impossible to successfully reconstruct the correct path in pointer execution tasks. In this Appendix section, we provide more details on a set of control experiments that were designed to ensure that this phenomenon did not play a role in our empirical evaluation.

As a preliminary step, we ablated in our experiments different strategies to structure our input components. However, we found that alternative designs for the tasks structure (e.g. LL-N-LL, where L represents a lower-case Latin letter and N a one-digit number) were always detrimental for the task accuracy compared to the current sequence design (i.e. LLNLL). To ensure that the input sequences of our pointer execution tasks were effectively tokenized as expected, we run some quantitative experiments with GPT-4's open-source tokenizer (`tiktoken`). Unfortunately, we could not find an open-source tokenizer for Gemini-Pro; the available token counter only provides limited information about the actual tokenization.

Analyzing the GPT-4 tokenizer results, we found that the digit delimiter ensures safe splitting within a word in $100\%$ of the samples, i.e. LLNLL always yields [LL, N, LL]. Additionally, we observed that $13.2\%$ of the 2-grams were split into two different tokens (e.g., `bq` is tokenized to `b` and `q`), which may affect the attention mechanism. As an additional experiment, we removed these 2-grams from the dataset. However, removing the "splittable" 2-grams did not result in an improvement of the performance (the accuracy decreased from 0.19 to 0.05 on PEN) when using the best-performing prompting technique with GPT-4 (Code Interpreter).

## D.2  Natural-PEN

We additionally designed a natural variation of the PEN task, dubbed Natural-PEN, where we replaced the synthetic 2-gram sequences used for the matching in the original tasks with 3-gram, in-distribution, natural English words. Natural-PEN inherits a similar structure of the components from the original PEN task, WNW, where W in this case are 3-characters English words. In practice, we filtered all valid 3-gram words from Scrabble[3] that were translated to a single token when using GPT-4's tokenizer. This gave us a set of 707 words (out of the original 1338 words). Similar to the experiment in the previous paragraph, we find that the GPT-4's performance using the best prompting techniques does not improve on this in-distribution, natural English words, yielding even a lower accuracy on Natural-PEN (Table D.13).

| Setup | Termination acc. | Match acc. | Task acc. |
|---|---|---|---|
| Few-shot CoT | 0.50 | 0.27 | 0.00 |
| Few-shot CoT, no traps | 0.20 | 0.29 | 0.00 |
| Code Interpreter | 0.05 | 0.10 | 0.05 |

Table D.13: Result of GPT-4 on the Natural-PEN task.

## D.3  Many-shot prompting

We additionally ablated the number of shot provided to the LLMs to perform in-context learning. We increased the number of shots from 8 examples up to 32 examples (64 did not fit into the 8k context window of the old GPT-4 version we are benchmarking). However, we did not observe any performance improvement, as reported in Table D.14.

---

[3]`https://scrabble.collinsdictionary.com/word-lists/three-letter-words-in-scrabble`

| | Termination acc. | Match acc. | Task acc. | Termination acc. | Match acc. | Task acc. |
|---|---|---|---|---|---|---|
| **Model** | | PEN | | | PERM | |
| GPT-4$_{8\text{ shots}}$ | 0.12 | 0.04 | 0.00 | 0.12 | 0.37 | 0.06 |
| GPT-4$_{32\text{ shots}}$ | 0.16 | 0.06 | 0.00 | 0.36 | 0.59 | 0.00 |
| Gemini$_{8\text{ shots}}$ | 0.05 | 0.03 | 0.00 | 0.08 | 0.04 | 0.04 |
| Gemini$_{32\text{ shots}}$ | 0.15 | 0.20 | 0.00 | 0.32 | 0.05 | 0.00 |

Table D.14: Ablation on the number of shots used to do in-context learning in PEN and PERM, for both GPT-4 and Gemini. In particular, the number of shots is increase from 8 examples to 32 examples. Despite increasing the partial metrics, providing more shots does not improve the overall task accuracy in both PEN (same, $0\%$) and PERM (the accuracy goes to $0\%$ for both models).

## D.4 Prompting OpenAI o1-preview

In this section, we include further results with the newly-released model from OpenAI, o1-preview[4], specifically trained to tackle complex tasks in science, coding, and math. The results are reported in Table D.15.

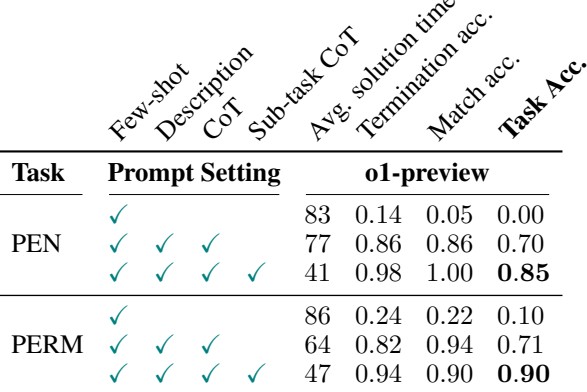

| Task | Few-shot | Description | CoT | Sub-task CoT | Avg. solution time (s) | Termination acc. | Match acc. | **Task Acc.** |
|---|---|---|---|---|---|---|---|---|
| | **Prompt Setting** | | | | **o1-preview** | | | |
| PEN | ✓ | | | | 83 | 0.14 | 0.05 | 0.00 |
| | ✓ | ✓ | ✓ | | 77 | 0.86 | 0.86 | 0.70 |
| | ✓ | ✓ | ✓ | ✓ | 41 | 0.98 | 1.00 | **0.85** |
| PERM | ✓ | | | | 86 | 0.24 | 0.22 | 0.10 |
| | ✓ | ✓ | ✓ | | 64 | 0.82 | 0.94 | 0.71 |
| | ✓ | ✓ | ✓ | ✓ | 47 | 0.94 | 0.90 | **0.90** |

Table D.15: Results of o1 with various prompt settings. While o1-preview cannot infer the task from examples only, it achieves much higher accuracy when given a description of how to perform the task. Our hand-crafted step-by-step solutions achieve higher accuracy than the solutions that o1 finds when asked for Chain of Thought (CoT) only.

---

[4]`https://openai.com/index/introducing-openai-o1-preview`

## D.5 Prompt examples

In this section, we include some exemplary prompts that were used in the PEN and PERM experiments with pre-trained LLMs. We start with the vanilla few-shot (8) prompt, shown in Table D.16.

---

**EXAMPLE**: xv ke vu7bh sb0fz xy5ih eo7sf ay7of xd3nj zs7bt eo1sf jn6yc xd5nj od3nk br2ny yc2pr ls5sg nv1zs sb5fz uy7vu sf1zv bh6ia sg5dg ux6oc zv4xd ya1yk br5ny wc4xy ke5fm jw1dx ny7sb wq2mm fz6eo nk2nv sf5zv pr3ya fz4eo yk0dk fm4br oc4wc nj0ls ih1uy di7fw mm2pq zv7xd of7wq nj4ls xv7gn ls6sg dx0ux vz7uc ah7od sg4dg sn2jw ae5ce ia7jn zw4ed bt5ay fm6br pq6kw ny3sb gn4ah ke0fm
**ANSWER**: ke ls6sg ke0fm sg4dg br2ny sf5zv sb5fz eo1sf fm6br xd3nj nj4ls fz6eo zv7xd ny3sb

**EXAMPLE**: wk qp ld2ki lt1ui te3ob et4mm ss1hi lt0ui ki3ay ad2dh zk1cc cl6lt zz3zs cl1lt hi3ct ap4ys sf1sz bc4yh ct4gf gn5ap mt5zd mm5gn sr1ne yh3et up0vl lu2cl no1ps ys3lu vm6us et6mm us4ha qp1ad zd6sf dh7bc ps3ld bc6yh sv5ch ad4dh qe4hd mm6gn ha0qe dh3bc vl3mt ys4lu ob2zk qp0ad wk7zz lu0cl ch7up ac2hh ne1te qo4qz cc2rp gx2yq xv3vm gn4ap rp2ss sd5gc zs6no yh4et tl7sv xs5jm sz0sr ww1rb ay6xv ap2ys
**ANSWER**: qp lu0cl cl1lt yh4et ys3lu bc6yh lt1ui ad2dh ap2ys gn4ap et6mm qp1ad dh3bc mm6gn

**EXAMPLE**: fy kn fz2jy st6zr wk4hz zo3st gz0nt ri2ru dt0rh ri4ru kq0lt st0zr pp3da ru2io lw0vk zo1st sx5xe ej2sj er1fz kn2ej fd2pp ej6sj mc7gs io0zo si4lp ns1jr xe6nb zn5ns vk4fd sj0im lt7lw kn4ej nb0lg sj2im lg0qi ru6io gs7yf zr3zn jy1wk im1ri fy5er ns7jr qi5dt io1zo yf7gz zn4ns nt6do im6ri lp6mc mk2ym do2kq zh3vh ox5si fk0sk hz3sx zr2zn
**ANSWER**: kn ns7jr kn2ej st6zr im1ri zo3st zr2zn ej2sj zn5ns sj2im ru6io io1zo ri4ru

**EXAMPLE**: xr ql hq6za co0gt yv6md sk5zi xr3kq zi0co zl4om mc7ha gy3ej xj0qp ud2hq ha0km yu7xa gt6nb rb4mi xj4qp li3ib km1cf lj2ek qp6sk ib4dv ha5km fc5gy gx1xd om7hb tx1mc al0lw gt4nb lw5fc xd6nj ek1tr nb5tx rz3yv ql1xj xu4it km0cf mi5vo gx5xd xa1in xd7nj vo6xz tx6mc dv4al co6gt kq2li sk6zi tr1zl cf7gx md0je mc4ha hh7rz cf2gx gp7yu zi2co je0gp nb0tx in2rb qp0sk it7hh lq4cl ej6lj ql0xj xz0ud sa2sc
**ANSWER**: ql zi0co sk6zi km1cf ha5km co6gt gt4nb xd6nj gx1xd xj0qp ql0xj qp6sk nb5tx cf7gx mc7ha tx1mc

**EXAMPLE**: bn xy fv6wc tn0jp sk2qb vy7mw od1hf ju6vs xf1kc xy6zn ue0dr zn7vy ya5fv ls4tn qb2km tn6jp yw6yn zn6vy pc5yw jp4vc af6rc mw7ls hf4om mw5ls kc4wt vv7ju hk3hn vy4mw hn3af vc3vv wt6pc vc6vv yn1sm ls2tn pe4pq jk3gb km7ep zv5gv om4vk mb7mz dr7hk jp0vc ep2pe is5ie sm2sk th7ba vz3ya ju1vs wc2ue xy2zn at5xd wq2wv bn5vz vv4ju vk3at gr0xr xd2xf up2vq
**ANSWER**: xy vv4ju ju1vs ls4tn tn0jp xy2zn zn7vy jp0vc vy4mw vc3vv mw7ls

**EXAMPLE**: kt nn fz2kv rv6wb kh6vu js4wk et7lx zk0sb ie3zb wb6mx vu5et nn7zx zb6zn pp3js kv4hk gt6rv bf4vr mx3zk sv5ok zk6sb gk1rx pp1js vg5kh wk5gt vr3fz zx2pp zu4ac nn3zx qe6rl gt4rv ac2tp rv5wb kt6zu sb4ic hk2xe sb1ic rx1xm mx6zk zn6vg az3vp xm4qe zx0pp dm1lt hl4li ok0ja wk3gt bc6bf tl2eb jn7dm ve3jl lt1bc tg5gp tp6sv wb4mx xe6ie ot2pj ja6gk js7wk
**ANSWER**: nn sb4ic nn3zx rv5wb wb4mx zk6sb wk3gt js7wk pp1js mx6zk zx0pp gt4rv

**EXAMPLE**: pq gg vs1dy cr4yf xt5kz gg1gn tg0fn gn4cr pq4jj tc1ts qh0ek yf2tc sk2al tc4ts jl3xi cr3yf ub1zh ts7lh tp2vb ex4px kz2kn gn3cr ez7up gg4gn dn2ez jm4ex ek1sz lh0eu kn5cg jm6ex jj5fp yf3tc il5wn eu3jm bd5fc ex7px wn5yw ir0xf fc3vs lh1eu fp0bd eu2jm fn6jl pa4zf zh4ku ep6jf dy4xt ts6lh vc5sk dv5nw sz2ub di7yp vb2dn tk0as ku0km mw7mx km0tp hi3pl bg4il nl1hv xi0bg kk2ao al1tg dx2ca up4vc ru0dc
**ANSWER**: gg tc1ts yf3tc eu2jm ex7px lh1eu cr4yf ts6lh gg1gn gn3cr jm6ex

**EXAMPLE**: if xe gw0is zl5jz if2mv wa5xs vs1st nz2wc ox0qs ko5xq xj4ue ko3xq hx7ym ca6zl is0bf xq6ed gs7ga xs0yg mv5ox ca5zl bg0bk xq1ed ry2gw rt7ca bf7lx xe1wa ym3vs xs7yg qs1ee wc6cb pe6ag ed3nz xn4dt zl7jz on6oc xe7wa ps0xj yg7ko st5ps cb2rt ag3xn wc2cb bk7hx rt2ca ee1ry cb4rt ra4on wa2xs lx7az ed0nz dt1ra be3hb rc4gs nz5wc az7rc yg0ko ue2pe or4fg
**ANSWER**: xe wa5xs ca5zl ko5xq wc6cb cb4rt rt7ca zl5jz xq6ed xe1wa ed0nz yg0ko nz5wc xs0yg

**YOUR QUESTION**: vq sk zg5gr ne7vc co7os hc6cv pq3ss dp4je vq3sx pu0dp wn2co vc6hc dw5vy ne6vc vy0oe gp6zm ka1rw vc1hc tz5mx je3hy ab2pq pu7dp oe3ka on5is lf1ju hy6gp gr5jp gp5zm os2dq sk6ne dq7cn on0is ju0qq cv2on mx1mb is0pu sx1wn cv1on rw4sz je6hy jp1tz dp0je io1lf is4pu ss5dw sk2ne sz4ij hc1cv qq6ab uk0af cn4zg hy2gp
Clearly mark your answer by writing 'Answer: <your answer>' as last line.

---

Table D.16: GPT4 and Gemini prompt for Pointer Execution's Neighbor (PEN) task with only few-shot examples of the task. The tasks are encoded so that every word start and word end of a word in the input are encoded as separate tokens, so a model can pattern-match the respective token to do the matching operation.

Then, we introduce a natural language description of the task, prepended before the examples of the task, as shown in Table D.17.

---

I give you a sequence of words. Each word has four characters plus a middle, words are separated by spaces. Start with the leftmost word. Output its neighbor. Then, match the last two characters of the current word (i.e. not the neighbor) to the word starting with those two characters. Again, output the neighbor. Do this until your current word (not the neighbor) has no match anymore.

**EXAMPLE**: xv ke vu7bh sb0fz xy5ih eo7sf ay7of xd3nj zs7bt eo1sf jn6yc xd5nj od3nk br2ny yc2pr ls5sg nv1zs sb5fz uy7vu sf1zv bh6ia sg5dg ux6oc zv4xd ya1yk br5ny wc4xy ke5fm jw1dx ny7sb wq2mm fz6eo nk2nv sf5zv pr3ya fz4eo yk0dk fm4br oc4wc nj0ls ih1uy di7fw mm2pq zv7xd of7wq nj4ls xv7gn ls6sg dx0ux vz7uc ah7od sg4dg sn2jw ae5ce ia7jn zw4ed bt5ay fm6br pq6kw ny3sb gn4ah ke0fm Answer: ke ls6sg ke0fm sg4dg br2ny sf5zv sb5fz eo1sf fm6br xd3nj nj4ls fz6eo zv7xd ny3sb

<7 MORE EXAMPLES>

**YOUR QUESTION**: vq sk zg5gr ne7vc co7os hc6cv pq3ss dp4je vq3sx pu0dp wn2co vc6hc dw5vy ne6vc vy0oe gp6zm ka1rw vc1hc tz5mx je3hy ab2pq pu7dp oe3ka on5is lf1ju hy6gp gr5jp gp5zm os2dq sk6ne dq7cn on0is ju0qq cv2on mx1mb is0pu sx1wn cv1on rw4sz je6hy jp1tz dp0je io1lf is4pu ss5dw sk2ne sz4ij hc1cv qq6ab uk0af cn4zg hy2gp

Clearly mark your answer by writing 'Answer: <your answer>' as last line.

---

Table D.17: GPT4 and Gemini prompt for Pointer Execution's Neighbor (PEN) task with few-shot examples of the task and a task description.

On top of this, we also add a request to reason step-by-step (CoT), as shown in Table D.18.

---

I give you a sequence of words. Each word has four characters plus a middle, words are separated by spaces. Start with the leftmost word. Output its neighbor. Then, match the last two characters of the current word (i.e. not the neighbor) to the word starting with those two characters. Again, output the neighbor. Do this until your current word (not the neighbor) has no match anymore.

**EXAMPLE**: xv ke vu7bh sb0fz xy5ih eo7sf ay7of xd3nj zs7bt eo1sf jn6yc xd5nj od3nk br2ny yc2pr ls5sg nv1zs sb5fz uy7vu sf1zv bh6ia sg5dg ux6oc zv4xd ya1yk br5ny wc4xy ke5fm jw1dx ny7sb wq2mm fz6eo nk2nv sf5zv pr3ya fz4eo yk0dk fm4br oc4wc nj0ls ih1uy di7fw mm2pq zv7xd of7wq nj4ls xv7gn ls6sg dx0ux vz7uc ah7od sg4dg sn2jw ae5ce ia7jn zw4ed bt5ay fm6br pq6kw ny3sb gn4ah ke0fm Answer: ke ls6sg ke0fm sg4dg br2ny sf5zv sb5fz eo1sf fm6br xd3nj nj4ls fz6eo zv7xd ny3sb

<7 MORE EXAMPLES>

**YOUR QUESTION**: vq sk zg5gr ne7vc co7os hc6cv pq3ss dp4je vq3sx pu0dp wn2co vc6hc dw5vy ne6vc vy0oe gp6zm ka1rw vc1hc tz5mx je3hy ab2pq pu7dp oe3ka on5is lf1ju hy6gp gr5jp gp5zm os2dq sk6ne dq7cn on0is ju0qq cv2on mx1mb is0pu sx1wn cv1on rw4sz je6hy jp1tz dp0je io1lf is4pu ss5dw sk2ne sz4ij hc1cv qq6ab uk0af cn4zg hy2gp

Reason step by step. Clearly mark your answer by writing 'Answer: <your answer>' as last line.

---

Table D.18: GPT4 and Gemini prompt for Pointer Execution's neighbor task with a task description, few-shot examples of the task, and a request to reason step-by-step.

We ablate the latter with a variation from Yasunaga et al. [28].

---

Your task is to tackle algorithmic problems. When presented with an algorithmic problem, recall relevant problems as examples. Afterward, proceed to solve the initial problem.

**# PROBLEM**: I give you a sequence of words. Each word has four characters plus a middle, words are separated by spaces. Start with the leftmost word. Output its neighbor. Then, match the last two characters of the current word (i.e. not the neighbor) to the word starting with those two characters. Again, output the neighbor. Do this until your current word (not the neighbor) has no match anymore.

**SEQUENCE**: oj ce rm3eo in7ds uy0az ds4ki hk5wo ee4bs hh5ve sl4in yl3om ce4qu wv4gq ds3ki mj7wv sl0in gq4kj zk2sl lu1nt in3ds nt0ye ki7ee oi7mm ki4ee sj7rg bs3fw mp6je ce2qu wo0mj qu6ei zy7ou ei2zk ve4mp qu7ei je0uy ee3bs eo5ow gv5sv ou1yl le3mz om0oa tq2zt rg5oi op3jm tn6hk pv1bp mm2tn zw0bd xq3lu zk6sl az6xq bs7fw oa0sj xv1js ow5zy yy2qb oj3hh ei4zk

**# INSTRUCTIONS**:
## Relevant Problems:
Recall three examples of algorithmic problems that are relevant to the initial problem. Your problems should be distinct from each other and from the initial problem (e.g., involving different numbers and names and instructions). For each problem:
- After "Q: ", describe the problem
- After "A: ", explain the solution and enclose the ultimate answer in \boxed{}.
## Solve the Initial Problem:
Q: Copy and paste the initial problem here.
A: Explain the solution and enclose the ultimate answer in \boxed{} here.

---

Table D.19: GPT4 and Gemini prompt for PEN with the prompting technique introduced by Yasunaga et al. [28].

We further improved the level of help in the prompt by providing few-shot chain-of-thought examples and sub-task few-shot chain-of-thought examples in Table D.20 and D.21, respectively.

---

I give you a sequence of words. Each word has four characters plus a middle, words are separated by spaces. Start with the leftmost word. Output its neighbor. Then, match the last two characters of the current word (i.e. not the neighbor) to the word starting with those two characters. Again, output the neighbor. Do this until your current word (not the neighbor) has no match anymore.

**EXAMPLE**: kt gu vv1vk xd4sp ra3kn fc6cd aq2oa co4wn wb7rr fl4tg zc1rk la7kb di1hn by3kj rk1fy zu6qf un6nb fe6fw fb4wb gu3fl ug5fb wn2kz hn5vv qy2cp nb7di ku6ah zh7qm ob1co rr6ti hi4ob fy1ra lc2sh oa6bw nk3pr ti2aq kz1nk kt2ug tg1hi op0zc ln0jz vk4op qk3ev bw4zh pr4ju

The leftmost word is "kt". Its right neighbor is "gu", so the first output word is "gu".
Now, we need to find a word that starts with "kt". The word is "kt2ug". Its right neighbor is "tg1hi", so the next output word is "tg1hi".
Now, we need to find a word that starts with "ug". The word is "ug5fb". Its right neighbor is "wn2kz", so the next output word is "wn2kz".
Now, we need to find a word that starts with "fb". The word is "fb4wb". Its right neighbor is "gu3fl", so the next output word is "gu3fl".
Now, we need to find a word that starts with "wb". The word is "wb7rr". Its right neighbor is "fl4tg", so the next output word is "fl4tg".
Now, we need to find a word that starts with "rr". The word is "rr6ti". Its right neighbor is "hi4ob", so the next output word is "hi4ob".
Now, we need to find a word that starts with "ti". The word is "ti2aq". Its right neighbor is "kz1nk", so the next output word is "kz1nk".
Now, we need to find a word that starts with "aq". The word is "aq2oa". Its right neighbor is "co4wn", so the next output word is "co4wn".
Now, we need to find a word that starts with "oa". The word is "oa6bw". Its right neighbor is "nk3pr", so the next output word is "nk3pr".
Now, we need to find a word that starts with "bw". The word is "bw4zh". Its right neighbor is "pr4ju", so the next output word is "pr4ju".
Now, we need to find a word that starts with "zh". The word is "zh7qm". Its right neighbor is "ob1co", so the next output word is "ob1co".
There is no word that starts with "qm", so we are done with the matching.
Therefore the answer is: "gu tg1hi wn2kz gu3fl fl4tg hi4ob kz1nk co4wn nk3pr pr4ju ob1co"

<7 MORE EXAMPLES> **YOUR QUESTION**: jm cq nu7nr th6cc du5ij ki7am nr0ad cq6th md4lk am4rw iu0lq bj7mu rp3ja rk1cs od3dm se3vv iw1hz si4rf dd5du mu4of wd0rt en2yt vw0al nb0ir qp6do ni0ff rt0ik sw2ji do5gn sh1bk kf0iu mm5mi ij2md cc4pq ad6dd rw0mm gn0rp ox5uc al2qp bo1fq hz3wd la1rb dm5vw de0ko jm7kf mi5bj wr1iw lu7lj lq7nu pq3ki ik6od da5gp
Reason step by step. Clearly mark your answer by writing 'Answer: <your answer>' as last line.

Table D.20: GPT4 and Gemini prompt for the Pointer Execution's neighbor task with few-shot chain of thought (Few-shot CoT) examples and a description.

---

I give you a sequence of words. Each word has four characters plus a middle, words are separated by spaces. Start with the leftmost word. Output its neighbor. Then, match the last two characters of the current word (i.e. not the neighbor) to the word starting with those two characters. Again, output the neighbor. Do this until your current word (not the neighbor) has no match anymore.

**EXAMPLE**: xa wo ri1ky kg3vt ky1pa mx6yl xa5fs yl6et cn0ry wo7du vx2ot tu7il ai1cn kg0vt ei7ex yl1et gk2qm mx4yl ot0ob fd2mx sn0ag fy7gn cw7hz du3kg ci5vx wo6du fs2ai et6fy pa6cw et2fy qm7gj du6kg xw6bn vt5fd bn7ei gn7tu zt7bl tu1il ag2xw mo7kd bl3gk vt6fd hz0sn nt4vr ob6ri ez2bj ry7dd fy6gn wi2zt fd1mx dd1wi gn3tu

First we find the matching sequence of words, then we find the neighbors.
The first word is "xa".
Now we need to find a word that starts with "xa". The word is "xa5fs".
Now we need to find a word that starts with "fs". The word is "fs2ai".
Now we need to find a word that starts with "ai". The word is "ai1cn".
Now we need to find a word that starts with "cn". The word is "cn0ry".
Now we need to find a word that starts with "ry". The word is "ry7dd".
Now we need to find a word that starts with "dd". The word is "dd1wi".
Now we need to find a word that starts with "wi". The word is "wi2zt".
Now we need to find a word that starts with "zt". The word is "zt7bl".
Now we need to find a word that starts with "bl". The word is "bl3gk".
Now we need to find a word that starts with "gk". The word is "gk2qm".
Now we need to find a word that starts with "qm". The word is "qm7gj".
There is no word that starts with "gj", so we are done with the matching.
Now we need to find the neighbors of the matched words.
The right neighbor of "xa" is "wo".
The right neighbor of "xa5fs" is "yl6et".
The right neighbor of "fs2ai" is "et6fy".
The right neighbor of "ai1cn" is "kg0vt".
The right neighbor of "cn0ry" is "wo7du".
The right neighbor of "ry7dd" is "fy6gn".
The right neighbor of "dd1wi" is "gn3tu".
The right neighbor of "wi2zt" is "fd1mx".
The right neighbor of "zt7bl" is "tu1il".
The right neighbor of "bl3gk" is "vt6fd".
The right neighbor of "gk2qm" is "mx4yl".
The right neighbor of "qm7gj" is "du6kg".
Therefore the answer is: "wo yl6et et6fy kg0vt wo7du fy6gn gn3tu fd1mx tu1il vt6fd mx4yl du6kg"

<7 MORE EXAMPLES>

**YOUR QUESTION**: zg kt tm2wb sj1re lg1at pb4sx la3ve qd3wa sc0iu my2sb wb6lg my1sb uq3ms zj1qd da2tm sx7sj xy6ny kt0dq ny7tn re4my os5xu sj2re tx5kp dq3zp yx5da sb2zj na3sc re5my gs5na sb4zj ly4qx qd0wa tn2ma dq6zp ma3gp zp3pb du2tx zj6qd fa4ly tz3cc vp3xy ry7ie ve6gs pb7sx do1ti ow5yy at6uq yk3wc gp7fa gg5eb ti5vp cv0dh ms6do bz0ut qx5bc jd3qb zg6os sx0sj iu6du kt6dq xu6la zp2pb bc4oz yb6vn oe4yx li5fz
Reason step by step. Clearly mark your answer by writing 'Answer: <your answer>' as last line.

Table D.21: GPT4 and Gemini prompt for the Pointer Execution's neighbor task with sub-task chain of thought (Sub-task CoT) examples and a description.

In Table D.22, we include the advanced sub-task few-shot CoT for the PERM task.

---

I give you a sequence of words. The last word (after the "|") is the word to start with. Now match match the last two characters of the current word to the word starting with those two characters. If this match was going to the left, i.e. the matched word is left of the current word in the sequence, increase a variable counting the number of left matchings. Do this until your current word has no match anymore.
Finally, output this sequence of words, in reverse order in the format word.x where x is the number of left matchings until the output word times the number of matchings until the output word. Example answer: abcd.4 efab.1 ghef.0

**EXAMPLE**: kp0ms gg0hy pk0tq go0ey mf0kp ms0jd hl0go vu0vu vl0gg bn0vl ar0pk tq0bn jd0hl hy0jm ey0oy oy0mf gy0do | ar0pk
First, let's enumerate the words:
1:kp0ms
2:gg0hy
3:pk0tq
4:go0ey
5:mf0kp
6:ms0jd
7:hl0go
8:vu0vu
9:vl0gg
10:bn0vl
11:ar0pk
12:tq0bn
13:jd0hl
14:hy0jm
15:ey0oy
16:oy0mf
17:gy0do
Starting with "ar0pk", let's match and calculate:
"ar0pk" matches with "pk0tq". The word "ar0pk" is 11th and "pk0tq" is 3th, so 1 left matches so far.
"pk0tq" matches with "tq0bn". The word "pk0tq" is 3th and "tq0bn" is 12th, so 1 left matches so far.
"tq0bn" matches with "bn0vl". The word "tq0bn" is 12th and "bn0vl" is 10th, so 2 left matches so far.
"bn0vl" matches with "vl0gg". The word "bn0vl" is 10th and "vl0gg" is 9th, so 3 left matches so far.
"vl0gg" matches with "gg0hy". The word "vl0gg" is 9th and "gg0hy" is 2th, so 4 left matches so far.
"gg0hy" matches with "hy0jm". The word "gg0hy" is 2th and "hy0jm" is 14th, so 4 left matches so far.
There are no further matches for "hy0jm", so we end the sequence here.

Finally, we calculate the number of left matches times the number of matches for each word and get:
ar0pk: 0*0=0
pk0tq: 1*1=1
tq0bn: 1*2=2
bn0vl: 2*3=6
vl0gg: 3*4=12
gg0hy: 4*5=20
hy0jm: 4*6=24
Thus, the answer is: "hy0jm.24 gg0hy.20 vl0gg.12 bn0vl.6 tq0bn.2 pk0tq.1 ar0pk.0".

<7 MORE EXAMPLES>

Your question: tb0tb sz0sp ls0tf vs0ch ek0in zg0ek ut0sl sp0sv um0ls rh0sz hy0kt sl0vs nh0ut in0hy tf0um sv0nh kt0zg nu0cb | rh0sz
Reason step by step. Clearly mark your answer by writing 'Answer: <your answer>' as last line.

---

Table D.22: GPT4 and Gemini prompt for Pointer Execution Reverse Multicount with description and sub-task few-shot chain of thought (Sub-task CoT). This chain-of-thought makes sure that the language model can easily execute all operations. In our experiments, for GPT4 to reliably determine whether in which order two words occur in a sequence needed enumeration.

In Table D.23, we include the prompt that was used for the code interepreter.

---

I give you a sequence of words. Each word has four characters plus a middle, words are separated by spaces. Start with the leftmost word. Output its neighbor. Then, match the last two characters of the current word (i.e. not the neighbor) to the word starting with those two characters. Again, output the neighbor. Do this until your current word (not the neighbor) has no match anymore.

**EXAMPLE**: bn oi fe6zt vj4ks yk3us fi6mk lo0ox yd5pt dc2rk mk3vj wj2fe wk0bm yo6kt ks5wk ox1yk wk6bm gc7kj oi1fi ah7gc ks0wk vq6ah gv7yd as4vq bm3gv vs3ho vj0ks kj0au yd6pt bn7dd mk2vj qg5lx pt4xv ho7lo oi5fi au7qg fi0mk zt3as nd4zd kt1vs gv0yd lx2dc fa0ut dd5ry pt3xv ry7yo bm0gv **ANSWER**: oi mk2vj pt3xv bm0gv ks5wk gv0yd vj0ks oi5fi yd5pt wk6bm fi6mk **YOUR QUESTION**: bc ek vq7ze pe7mu vu5pf nu1sb zg2uf vd5vn ep3qt ek2oj ng6ep ig7is bc4sx is2nu sx3lq pe2mu lq4oo sb1vd wh0lw oj7ax qb2vq mu4xm hq6eu nu0sb pf3qb ax2pe yh7bm sb0vd gz5vu vd1vn fa5sj oj0ax eu0zg ek4oj ze3gx xm3ig oo5hq mu7xm bp7gz is7nu lw3ng vf7ko sj3jq xm4ig xp0fa ax6pe gx3wh qc6tc uf5xp ig3is qt5yh rs5lk

Reason step by step. Then, use the code interpreter to solve the task. Clearly mark your answer by writing 'Answer: <your answer>' as last line.

---

Table D.23: GPT4 prompt for the Pointer Execution's neighbor task with code interpreter.

I give you a sequence of words. The last word (after the "|") is the word to start with. Now match match the last two characters of the current word to the word starting with those two characters. If this match was going to the left, i.e. the matched word is left of the current word in the sequence, increase a variable counting the number of left matchings. Do this until your current word has no match anymore. Finally, output this sequence of words, in reverse order in the format word.x where x is the number of left matchings until the output word times the number of matchings until the output word. Example answer: abcd.4 efab.1 ghef.0

**EXAMPLE**: ud0xg wp0mr yy0uo xg0yy sr0mw pg0yg oq0zt mw0oq uo0bt ep0ep rs0av bt0oi oi0ud mr0pg oc0wp av0oc tz0tb yg0tz tb0rs vk0sx | sr0mw
Starting with "sr0mw", let's match and calculate:
"sr0mw" matches with "mw0oq". No left match, 0 left matches so far.
"mw0oq" matches with "oq0zt". This is a left match, 1 left matches so far.
There are no further matches for "oq0zt", so we end the sequence here.
Reversing the sequence and formatting it as per the instructions, we get:
oq0zt.2
mw0oq.0
sr0mw.0
Thus, the answer is: "oq0zt.2 mw0oq.0 sr0mw.0".

**EXAMPLE**: vs0ep xv0tx me0xt xx0ds ds0re re0rm tx0tg sh0vs xt0bw pf0ss gh0sh ep0me bw0xv rm0gh ss0pf kh0of | xx0ds
Starting with "xx0ds", let's match and calculate:
"xx0ds" matches with "ds0re". No left match, 0 left matches so far.
"ds0re" matches with "re0rm". No left match, 0 left matches so far.
"re0rm" matches with "rm0gh". No left match, 0 left matches so far.
"rm0gh" matches with "gh0sh". This is a left match, 1 left matches so far.
"gh0sh" matches with "sh0vs". This is a left match, 2 left matches so far.
"sh0vs" matches with "vs0ep". This is a left match, 3 left matches so far.
"vs0ep" matches with "ep0me". No left match, 3 left matches so far.
"ep0me" matches with "me0xt". This is a left match, 4 left matches so far.
"me0xt" matches with "xt0bw". No left match, 4 left matches so far.
"xt0bw" matches with "bw0xv". No left match, 4 left matches so far.
"bw0xv" matches with "xv0tx". This is a left match, 5 left matches so far.
"xv0tx" matches with "tx0tg". No left match, 5 left matches so far.
There are no further matches for "tx0tg", so we end the sequence here.
Reversing the sequence and formatting it as per the instructions, we get:
tx0tg.60
xv0tx.55
bw0xv.40
xt0bw.36
me0xt.32
ep0me.21
vs0ep.18
sh0vs.10
gh0sh.4
rm0gh.0
re0rm.0
ds0re.0
xx0ds.0

Thus, the answer is: "tx0tg.60 xv0tx.55 bw0xv.40 xt0bw.36 me0xt.32 ep0me.21 vs0ep.18 sh0vs.10 gh0sh.4 rm0gh.0 re0rm.0 ds0re.0 xx0ds.0".

<7 MORE EXAMPLES>

**YOUR QUESTION**: lw0ws gs0gs eq0eq yz0pt ws0lw um0qd ea0ea tf0um df0df uo0tf fl0uo pt0yz po0ec | fl0uo
Reason step by step, Clearly mark your answer by writing 'Answer: <your answer>' as last line.

Table D.24: GPT4 and Gemini prompt for Pointer Execution Reverse Multicount with description and fewshot-chain-of-thought we found to be similar to the chain of thought the model chooses naturally. The 0s in the input words are inserted to make sure, GPT-4 encodes each word start and word end as one token and therefore can match more easily.

