# OpenReview forum: "Limits of Transformer Language Models on Learning to Compose Algorithms"
_NeurIPS.cc/2024/Conference — NeurIPS 2024 poster_

### Official Review · Reviewer_Fa35 · 2024-07-07

**Soundness:** 2
**Presentation:** 1
**Contribution:** 2
**Rating:** 4
**Confidence:** 3

**Summary:**

This paper studies whether transformers can efficiently learn compositional discrete tasks. In particular, the paper introduces two new tasks: pointer execution neighbor and pointer execution reverse multicount as well as using multiplication and highest subsequence sum from prior work. First, small models are trained from scratch, showing substantially slower learning on the composition than on the subtasks. Next, API models are prompted to solve the same tasks and perform somewhat poorly. Some theory is also provided showing how models that memorize can struggle to learn compositions efficiently.

**Strengths:**

1. The paper proposes an interesting question as to whether we can determine whether a language model has some higher level concept of task composition that allows it to learn compositions of previously learned tasks efficiently.

2. The paper includes a nice theoretical result via a complexity theory reduction that shows how composition is hard if we assume stylized models that memorize the training data.

**Weaknesses:**

1. H1 as written cannot be disproven empirically since the "constant" could just be larger than those tested. It seems in the experiments "constant" means 100. If that is what is meant, then just say so in the hypothesis.

2. It is not clear if the notion of "sub-task" is somehow atomic and unique. This makes hypothesis H2 and H3 somewhat ill-defined too. It is possible that there are different sub-tasks (and perhaps many more of them) that better track how the model actually learns. Just because we can posit one way to compositionally solve the task does not mean that the model will learn that way (or even that it can necessarily represent that composition).

3. It is not clear why the new tasks are necessary or what specifically they add over prior, simpler tasks. There needs to be more justification of the somewhat complicated tasks to explain why they are necessary. Generally the presentation of these tasks and the results was unclear and could use improvement to make it more visually clear how matches and transitions are meant to happen in the tasks and more precisely what all the baselines are doing in the experiments.

4. It is not clear why one would expect an untrained small (150m) language model to somehow be able to compose subtasks without being trained to perform composition. As such, the results that the composition does not just arise and indeed takes longer to learn is not surprising.

5. I am somewhat worried that the way the strings representing the nodes are designed is interacting badly with the tokenizers in the API experiments. These are clearly "out of distribution" types of words and they may be tokenized in ways that make it very difficult to solve the task. Did you do any analysis of how these strings get tokenized? The tokenizers are publicly available. Also, it is difficult to fit this section into the story of the paper since there is no comparison to the learning of the subtasks.

**Questions:**

See weaknesses.

**Limitations:**

Limitations of not solving the issues raised are addressed in the paper.

---

> ### Author Rebuttal · Authors · 2024-08-07
>
> Thank you for your insightful comments and observations.
>
> ---
>
> **(W1)** As shown in our results, H1 can be fixed to a very large range, and 100 was chosen for illustrational purposes; what is crucial is that we assume it is empirically much lower than H2. This means that H1 is defined to make the hypothesis more accessible and actionable, and it should only depict an upper bound of "a few demonstrations" i.e., we got inspired by what one would usually give in a few-shot and took an upper bound. We also refer to the PAC learning theorem, where the logarithmic factor shows that with very few samples, a large hypothesis space can be reduced, making a constant assumption on limited-complexity tasks reasonable in practice.
>
> ---
>
> **(W2)** Thanks for the critique, it is indeed an important aspect that was not sufficiently clarified in the manuscript. In the introduced tasks, every primitive roughly corresponds to a basic block of operations in the equivalent procedural programming language and can be considered an atomic unit. To visualize the atomicity of the primitives within the tasks, we included pseudocode that shows how the introduced tasks (PEN and PERM) can be decomposed and solved with these primitives by a language model trained on next-token prediction in the additional 1-page PDF note.
>
> The definition of these sub-tasks is, per se, not unique. However, our definition of sub-tasks is _minimal_, as every primitive is independently observed in only one task (as described in Section 2.2), and the primitives are atomic. In addition, we argue that our observations are independent of the optimality and uniqueness of primitives and sub-tasks. The inefficiency observed is _relative_ to the given set of primitives and not defined in absolute terms. Finally, we cannot completely exclude the existence of an "optimal" set of primitives that could be easily re-combined by the model to learn compositional tasks efficiently. However, this possibility is deemed unrealistic by some of our experimental results. For instance, in our de-compositionality experiments (Section 4.2 and Tables B8, B9) we see that the model learns the sub-tasks much faster when pre-trained on the composition task. We speculate that this provides evidence that the model learns mechanisms similar to our primitives and that the learning inefficiency is, hence, mostly imputable to the difficulty of the model to recombine the primitives (which would not be solved by a better set of primitives).
>
> ---
>
> **(W3)** Thank you very much for your question. We tried to improve the exposition of the tasks and their motivation in our rebuttal to the reviewer teuc. In addition, we also improved Figure 1, please refer to the additional 1-page PDF note.
>
> ---
>
> **(W4)** This is a valid concern. Firstly, we argue that experimenting with small, un-trained models can still be interesting to test whether the compositionality can emerge naturally or not (as this result could be then potentially translated to real-world practical application) and provide insights on the intrinsic limitations of Transformer-based models on these tasks. Then, we highlight that in our experiments we also incorporated a certain amount of inductive bias toward compositional solution. For example:
> - _In-context learning_. We prompted GPT-4 and Gemini-Pro extensively and demonstrated that they did not learn to compose their primitives successfully enough to even perform the tasks when given a clear description and a large number of examples. Please find the new results with many-shot prompting in the general response.
> - _Training the model from scratch_. Note that we provided hints to the model such that learning how to compose learned primitive sub-tasks is made easier. We do this by introducing the PEV (Pointer-Execution Verbose) to the list of sub-tasks, which implicitly shows to the model the order in which the other subtasks need to be applied to solve PEN. For results with different architectures, please see "ablations with different architectures" in our general response.
> - _Fine-tuning of larger pre-trained models_. Inspired by your comment, we have designed a new set of experiments to fine-tune larger pre-trained models on our algorithmic tasks, to validate whether the bigger size and/or the pre-train knowledge would affect our observations. Unfortunately, this kind of training takes a considerable amount of time, but we hope to be able to provide practical results before the end of the discussion period.
>
> ---
>
> **(W5)** Thank you for raising potential concerns regarding tokenizations. We already searched through different task designs to improve the performance of GPT-4 and Gemini-Pro. The reviews motivated us to conduct further investigations and quantification primarily on the openly available GPT-4 tokenizer. Our findings can be summarized as follows (see the general response for a detailed discussion):
> - We ablated different task designs in the initial submission; however, we found that the current sequence design (e.g., "ab1cd") performs best among other alternatives (such as "ab-1-cd").
> - As an additional analysis, we tested the tokenization of GPT-4's open-source tokenizer (tiktoken). We found that the digit delimiter is always tokenized separately (e.g., "ab1cd" yields "ab", "1", and "cd"); hence, it does not have a detrimental effect on the attention mechanism. We found that some of the 2-grams were split into 2 tokens; however, removing them from the dataset did not improve the overall task accuracy with the best-performing prompting technique.
> - We designed a new natural variation of PEN, dubbed Natural-PEN, where we replaced the synthetic 2-gram sequences with 3-gram, in-distribution, natural English words. Overall, we found that the GPT-4's performance does not improve on this in-distribution, natural English words, yielding even a lower accuracy on Natural-PEN.

---

> > ### Comment · Reviewer_Fa35 · 2024-08-09
> >
> > Thanks for the response and clarifications.
> >
> > 1. Using ideas like "a few demonstrations" is just not formal. And dressing this up as a formal hypothesis just seems to be misdirection when the experiments just choose some arbitrary number out of thin air. This is honestly a small part of the paper, and does not majorly impact my opinion, but it is bad practice.
> >
> > 2. Thanks for the discussion here. I agree that it is a tricky issue for sure, but I am still not totally convinced that these must be the ways to decompose the tasks or that it is conclusive that the model is actually learning them in the way presented.
> >
> > 3. Thanks for the figure. It is still not clear to me why the new task definitions are necessary.
> >
> > 4. Thanks again for the clarification. Indeed adding the hints that get closer to showing how to do composition is maybe helpful. I guess my point was that maybe these synthetic tasks are missing the diversity of natural data that may have more examples of *how* to do composition in a general sense. And just training on narrow subtasks and expecting composition to arise seems unrealistic (and also not being suggested by people in the literature as being something that would occur, to my knowledge)
> >
> > 5. Thanks for these extra experiments and investigation. This does seem to be better evidence that the tokenization is not a major issue.
> >
> > I will increase my score to a 4 to reflect the partial resolution of some of the issues I raised, but I think the remaining issues still leave the paper below the bar for acceptance in my mind, especially with several substantial changes to the paper being suggested.

---

> > > ### Author Response · Authors · 2024-08-11
> > >
> > > Thank you for raising your score, for the additional comments on our work, and for your involvement in the review process, we really appreciate it.
> > >
> > > **(1)** We agree that the definition right now lacks formality and that this should be addressed. We propose a simple variation of our initial hypothesis, which does not affect either our observations or our experimental setup, yet fixes the weak definition of H1.
> > > - H1. A Transformer language model learns a compositional task with fewer samples than those required to learn its most difficult sub-task.
> > > - H2. A Transformer language model learns a compositional task with fewer samples than the sum of the samples needed to learn every sub-task.
> > > - H3. A Transformer language model learns a compositional task with approximately (the same order of magnitude) as many samples as the sum of the samples needed to learn every sub-task.
> > > - H4. A Transformer language model needs more data samples than in H3.
> > >
> > > **(2)** It is indeed hard, given the limited explainability of the internal mechanisms of Transformer models, to establish whether or not the model is effectively learning certain operations. However, the experiments on the task decomposition represent in our view a solid clue that LLaMa learns internal operations related to the set of primitives that we define (since pre-training on the compositional task results in a faster learning of the individual sub-tasks). Additionally, we argue that this experiment also proves the validity of our choice of primitives: when trained only on the compositional task, the model autonomously converges to operations that are related to them and can effectively compose them to solve the task. Hence, we believe that assuming that the model should be able to leverage the primitives in the given sub-tasks to learn the compositional task does not seem too far-fetched.
> > >
> > > This, in some sense, links back to some of your remarks in the initial review which we might now have completely addressed in the previous response.
> > > > Just because we can posit one way to compositionally solve the task does not mean that the model will learn that way (or even that it can necessarily represent that composition)
> > > In practice, the experiments on de-compositionality show exactly these two points.
> > >
> > > To sum up the rather long discussion on the topic, these were the main important clarifications and arguments.
> > > - _atomicity_: the sub-tasks are effectively atomic units roughly corresponding to a basic block of operations in the equivalent procedural programming language;
> > > - _uniqueness_: the sub-tasks are not unique because we could consider an arbitrary set of sub-tasks composed of primitives or combinations of primitives that make each one of them observable. However, we choose the minimal set of sub-tasks that makes every operation observable.
> > > - _validity of the primitive set_: experiments on task de-composition corroborates the hypothesis that the considered primitives can be learned and be effectively composed when training only on the full compositional task. This represents a validation of our choice of primitives and our experimental setup in general.
> > > - _relativity_: the measured inefficiency is relative to our specific definition of sub-tasks and primitives, as correctly pointed out. However, considering all the points above, we speculate that similar results could be measured also for alternative definitions.
> > >
> > > **(3)** We included an extensive rebuttal on the nature of the introduced tasks and why they are a meaningful contribution to the community in general in the response to reviewer teuc. We include a summary of our main arguments in the following list.
> > > - First, a main driver for the design of our new tasks was the absence in the landscape of research on compositional learning of algorithmic tasks based on the pointer execution (PE), a synthetic evaluation task first introduced by Bengio et al. [20] to test the generalization capabilities of deep learning models. This task limits the number of confounders in the data and, therefore, reduces (if not annihilates) the number of shortcut solutions that the model can find. In other words, pointer execution tasks force the model to learn an algorithmic (general) solution by design and not rely on any possible statistical shortcut that might compromise the validity of the observations.
> > > - Second, PEN and PERM are particularly suitable for benchmarking compositionality due to their algorithmic nature. We can decompose them into single atomic operations (referred to as primitives in the text), which can make learning the task easier if the model can identify and leverage their compositional nature.
> > > - Finally, our tasks represent a unique example in the sense that the failure of GPT-4 and Gemini-Pro is very apparent despite extreme help, e.g., providing multi-shot examples and showcasing limited compositionality of current SOTA models better than any task we are aware of.

---

> > > > ### Author Response · Authors · 2024-08-11
> > > >
> > > > **(4)** We agree with you that the synthetic tasks considered might be missing the diversity of natural data that may have more examples of how to do composition in a general sense. This is intentional. We chose to provide the model only with a minimal set of sub-tasks and avoid this kind of redundancy.
> > > >
> > > > Firstly, because in these settings the risk of learning spurious correlations and other possible confounding phenomena (such as the unobserved local structures anomalies pointed out by DGs4) would increase, possibly compromising our results.
> > > >
> > > > Secondly, because increasing the set of observed combinations of primitives beyond the minimal set is itself an inefficiency. We could also have provided many more combinations of sub-tasks to artificially "simulate" this regime, but that would have added an additional dimension to our investigation that is, however, inefficient by design.
> > > >
> > > > Finally, we cannot see any reason why a model that does not perform well on simple, well-defined synthetic tasks would perform better in a more natural and complex setting. Instead, the only solution that we see to the learning inefficiency consists in adding more inductive bias into the model, as was the case for example in our experiments with the additional prediction heads (experiments with the _auxiliary loss_ in Section 4.1).
> > > >
> > > > **(5)** We are happy to hear that our additional results on tokenization were satisfying. Thanks again for suggesting to investigate more closely into this aspect of our experiments.
> > > >
> > > > Finally, we understand your concern regarding the excessive changes to our initial submission. However, most of the proposed changes would be added in the form of Appendix material to address possible objections and secondary aspects of our study (e.g., tokenization, other models, etc.). The changes in the main document would concern mostly the presentation of the tasks, on which many reviewers (rightfully) had some concerns, while the main message of the paper and the main empirical results would remain consistent with the original submission.

---

### Official Review · Reviewer_teuc · 2024-07-12

**Soundness:** 3
**Presentation:** 3
**Contribution:** 2
**Rating:** 6
**Confidence:** 2

**Summary:**

This paper focuses on analyzing the transformer language models' learning and transferability on compositional discrete tasks. Specifically, it has four hypothesis, and the author studies for a variety of language models, whether does these hypothesis hold.
H1. An LLM can learn to perform a compositional task with constant number of datapoints.
H2. An LLM can learn to perform a compositional task, given as many sample as the most difficult sub-task required.
H3. An LLM can learn to perform a compositional task, given the data samples of relearning all sub-tasks for learning the composition.
H4. An LLM can learn to perform a compositional task, given more data samples in H3.
The authors introduces a new benchmark for creating systematic sub-tasks and testing compositionally.
With LLaMA model, H4 holds; with both GPT-4 and Gemini, using H1 (prompting) fails to perform the tasks, or multi-round code generation with COT technique.

**Strengths:**

Originality: 3.5 / 5

This paper examines how the number of datapoint samples affects the learning of compositional tasks in existing transformer-based large language models (LLMs). The authors created a new, challenging compositional dataset based on computation graphs and demonstrated that learning compositional tasks with LLMs is highly data-inefficient. While the BIG-bench paper indicates the insufficiency of reasoning and compositional abilities in LLMs, this paper innovatively provides a concrete, quantitative, and extensive study on the extent of this insufficiency.

Quality: 3.5/5

The empirical study is pretty extensive with both LLaMA, GPT-4 and Gemini. There are multiple prompting techniques adopted with GPT-4 and Gemini, all of them fails to generate a reliable result. There are also very interesting theotrical proofs in the appendix to bolster the authors' claims.

Clarity: 3/5

Figure 1 is hard to understand just by staring at the graph. For each task, it only provides one example which is non-trivial at all. One can hardly figure out its ground truth program for each example, and whether in a task of PE, is the underlying program the same across all the datapoints. I believe a descriptive caption by the side of each task is necessary. For example, PE refers to a program that takes a sequence of words and returns a list of words all colored green, where the first output word matches the first input word, and any subsequent output word starts with the last two characters of the previous word. However, the figures and tables in the experimental section are pretty clear and helpful to understand.

Significance: 2.5/5

Understanding the problem of the data inefficiency in transformer based LLMs is important to the community which focuses on data efficiency and reasoning, such as neuro-symbolic community.

**Weaknesses:**

As stated in the strengths above. One of the main issue is the clarity issue of the tasks. Besides "what is the task", I also want to understand "why these two tasks are needed". What do PEN and PERM these two datasets bring?

**Questions:**

Q1. Is the PEN dataset only corresponding to one underlying program?
Q2. What are the insights of PEN and PERM these two datasets?

**Limitations:**

I believe the stated limitation regarding addressing weaknesses in the LLM is not appropriate for this specific paper. Instead, the limitation should focus on the choice of the compositional dataset.

---

> ### Author Rebuttal · Authors · 2024-08-07
>
> Thank you for your positive feedback, insightful comments, and observations.
>
> **PEN and PERM: clarification and motivation**
> To better understand the PEN and PERM tasks, we start the exposition by explaining the original Pointer Execution (PE) task using our encoding scheme.
>
> The PE task is similar to linked list traversal or pointer chasing. The inputs are words delimited by whitespaces, where each word consists of an address, a delimiter, and a value. In our setup, addresses and values are each encoded using two letters from the English alphabet, and the delimiter is a single digit between 0 and 9. For example, uk7bh has "uk" as the address, 7 as the delimiter, and "bh" as the value. The first word in the input sequence is a single address, which points to a unique word in the sequence, starting the linked list traversal process. The sequence is traversed by reading out the value of the current word and matching it to the unique word whose address equals the current value. The traversal ends once there is no word whose address equals the value of the current word.
>
> For example, given the input sequence "aa bc1yu aa5bc op9mn", the correct word traversal order is "aa aa5bc bc1yu".
>
> This type of task is especially suitable for our study, as it limits the number of confounders in the data and, therefore, reduces (if not annihilates) the number of shortcut solutions that the model can find. In other words, PE tasks force the model to learn an algorithmic (general) solution by design and not rely on any possible statistical shortcut that might compromise the validity of the observations.
>
> However, algorithmic tasks based on pointer chasing have so far been absent from the landscape of research on compositional learning. This is in large part due to the difficulty in expressing the PVR task using any primitive procedure except ``match``, which extracts the value of the word currently processed and finds the word whose address matches the value. Hence, our PEN and PERM tasks emerged from the need for a set of tasks that keep the beneficial properties of pointer-chasing tasks and can simultaneously be expressed as a composition of a set of simple primitives.
>
> The algorithmic primitives we introduce in the PEN task are ``left`` and ``right``, which require the model to find the word to the left resp. to the right of the word that is currently being processed.
>
> The input sequence is also modified and now consists of two sequences with interleaved elements. The first sequence is a valid PE sequence, whereas the second sequence is reminiscent of a PE sequence, but the values and addresses are not restricted to be unique. The two sequences have no common addresses or values. As an example, take the two sequences "ab ab8cd cd6ef" and "gh gh9kl gh6ij". The first one is a valid PE sequence, and the second one has the repeated address "gh". The corresponding PEN sequence is "ab gh ab8cd gh9kl cd6ef gh6ij". The task consists of traversing the pointers of the first of the two interleaved sequences while, at each traversal step, outputting the word to the right of the word currently traversed (the neighbor).
>
> The fact that the model needs to output the neighbor combined with the neighbor's value, possibly having several matching addresses in the sequence, makes PEN a challenging compositional task, which can still be solved using only a few algorithmic primitives.
>
> The PERM task is also based on the PE task, with a slightly different encoding of the sequence elements. The elements now consist of four letters from the English alphabet, to be interpreted as pairs of addresses and values, each encoded using two characters. The first element in the pointer traversal is explicitly provided.
>
> PERM builds on top of the PE task by requesting that the sequence of words in the pointer chasing problem be output in reverse order, and it additionally involves keeping track of the length of the chain of pointers as well as the pointer orientation, i.e., does it point to the left or to the right of the current word.
>
> Let us start with the following example sequence: "aabb eeff ccdd bbcc ddee", with the initial word being "aabb". Following the sequence of pointers starting from "aabb" results in the following sequence: "aabb bbcc ccdd ddee eeff". Each element in the resulting sequence has two numerical values associated with it, these being the length of the preceding pointer chain ('count') as well as the number of 'left'-pointers in the chain preceding the current element ('count left'). In our example here, the relevant sequences and numerical values are represented below:
>
> Input sequence:             "aabb eeff ccdd bbcc ddee"
>
> Pointer chasing result:     "aabb bbcc ccdd ddee eeff"
>
> Count:                        0    1    2    3    4
>
> Count left:                   0    0    1    1    2
>
> Each step in the pointer-chasing procedure increases the "count" by one.
>
> "bbcc" -> "ccdd" is the first pointer oriented to the left, and so it increases "count left" from 0 to 1. "ddee" -> "eeff" is the second pointer oriented to the left, and it increases "count left" from 1 to 2.
>
> The final result is obtained by outputting the pointer chasing result in reverse order and associating with each word in the output sequence a number equal to "count" multiplied by "count left" for that word. This results in the following output:
>
> Input sequence:  'aabb   eeff    ccdd    bbcc    ddee'
>
> PERM result:     'aabb.0 bbcc.0  ccdd.2  ddee.3  eeff.8'
>
> ---
>
> **(Q1)** We cannot guarantee that our decomposition of the PEN task into the primitives "match", "left" and "right" is unique. A different set of primitives will lead to a different decomposition. However, given our set of very simple primitives, it is highly unlikely that there exists a decomposition different from what we provide in the PDF.
>
> ---
>
> **(Q2)** We refer to our answer Q1 of reviewer qmuR.

---

> ### Comment · Reviewer_teuc · 2024-08-11
>
> Your response has clarified my concerns. I have raised my score from 4 to 6. I really hope these clarifications can go into the revised version of the paper to improve the clarity and presentation.

---

> > ### Author Response · Authors · 2024-08-11
> >
> > Thank you very much for raising your score, we are glad that our response clarified your concerns.
> > We will indeed incorporate all this material in the revised version of the manuscript.
> > Once again, thanks for highlighting these issues and contributing to the improvement of our work.

---

### Official Review · Reviewer_DGs4 · 2024-07-13

**Soundness:** 4
**Presentation:** 3
**Contribution:** 3
**Rating:** 7
**Confidence:** 4

**Summary:**

This paper evaluates the compositional learning abilities of Transformer-based models with LLaMA-like architecture on tasks requiring the composition of several discrete sub-tasks. To this end, the paper reuses two existing compositional algorithmic tasks and introduces two new ones, focusing on how many samples are needed for models to learn to compose the sub-tasks compared to the sample efficiency of learning the sub-tasks themselves. The study measures the efficiency of models when trained from scratch and the effectiveness of prompting the pretrained language models GPT-4 and Gemini. The experiments suggest that hypotheses that compositional learning requires no more samples than the sum of samples needed for each subtasks should be rejected. The paper also performs few-shot prompting with GPT-4 and Gemini with different prompting techniques to investigate their ability to learn to compose or decompose algorithms in-context and find that they are unreliable for executing sub-tasks or correcting errors in multi-round code generation. Finally, the paper uses complexity theory to support these findings, suggesting that when training feedforward models to memorize information with gradient descent, the sample inefficiency is inevitable.

**Strengths:**

1. Aside from achieving state-of-the-art performance on many academic benchmarks, transformer-based language models are the undisputed workhorse for numerous real-world applications. However, their training does not necessitate compositional learning explicitly, while many of the tasks they are tasked at solving do require such capability. As such, understanding the limits and requirements for these models to learn to compose independent skills is key to drive our understanding of these foundational models and to improve them.
2. The analyzed tasks in the paper are very well defined to verify that a model that learns a task must know how to perform the subtasks, and that given capability to solve the subtasks, a model must only learn to compose these abilities to solve the task itself. Creating such settings is not trivial, and goes a long way to enhance our understanding of the compositional learning abilities of transformer models.
3. The paper provides a very thorough literature review and contextualizes the work around prior work very well.
4. The presentation of the paper is generally very nice, making a technical read easier and fluent.

**Weaknesses:**

1. The authors correctly identify tokenization as a possible negative confounder in the defined testbed, and thus use character-based tokenization for the training experiments. However, the same care is not taken when investigating the abilities of GPT4 and gemini to perform in-context-learning. Namely, given the highly synthetic nature of the inputs, it is highly possible that both the out-of-domain distribution of these inputs (deviating from natural texts) as well as differences in how inputs are tokenized (for example, one key can be tokenized with a single token, another by three tokens, and a third be tokenized along with parts of the corresponding value) confounds and skew the results, hindering their usefulness.
2. Moreover, while the authors study many prompting techniques to evaluate GPT4 and Gemini, they use a constant 8-shot prompting. It is known that these models can benefit greatly from increased number of demonstrations, and has been shown that for out-of-domain tasks, one has to prompt the model with significantly more than 8 prompts to perform well (e.g. Levy et al., 2022, Bertsch et al. 2024, Agarwal et al. 2024)
3. The proposed testbed is very well defined, but a single transformer based model is being studied. Understanding the contextualization of the results given difference in the model and tasks properties (for example width-depth ratio, scaling behavior with respect to parameter count or effect length of the inputs) would be very beneficial.ֿ

**Questions:**

1. Can you imagine similar experiments with more natural compositional tasks that can be learnt, and then used to benchmark SOTA LLMs in the settings they were designed and trained for? For example, do you think it is possible to use something similar to the unobserved local structures as proposed by Bogin et al. (2022) to create such settings?
2. Can you try repeating the experiments with clear instruction for GPT4/Gemini but using as keys only tokens in their respective vocabularies separated by e.g ‘-‘ so that the tokenization and highly synthetic nature of the task would have less detrimental effect on the results?
3. What are the exact specifications of the trained model? You mentioned it is 150M parameters, does that mean it is a 12-layer decoder only model? What was the training procedure in terms of hyperparameters? Do you see any scaling behaviors a-la Kaplan et al., 2020? For example, does using a larger model decrease the amount of samples needed? Also, does the depth of the model affect the ability to learn to compose subtasks?
4. Presentation suggestion - in a few places in the text, it would be very helpful for the reader to be able to read the descriptions if they were coupled with examples.
    1. In section 2.2 you define sub-tasks in the computation graphs. I think including a small demonstration of such a computational graph, along with toy examples of sub-tasks and tasks would go a long way to make this section clearer and improve the smoothness of the reading.
    2. In Section 3 you define and explain the different tasks and subtasks of your testbed. While Figure 1 is very nice and contributes a lot, it does not explicitly show the different sub-tasks or the procedures involved in deriving the output from each input, and in some cases (for example the counts in PERM) it may take time for the reader to understand what are the exact tasks. I think a more explicit demonstration of the procedure would be very helpful. It can either be a step-by-step demonstration of the procedure for each task (added in the appendix for brevity), or even a textual derivation of the procedure applied to clarify the operations being performed at every step.
    3. In section 4, it will be very useful to add a table with an example input and output used for each task and their statistics (e.g. length, histogram on number of neighbor steps needed etc). If the inputs and outputs in Figure 1 are representative, you can also say that directly and point there.

**Limitations:**

The authors discuss some limitations, but do not directly address the generalizability of their findings to natural tasks subtasks composition.

---

> ### Author Rebuttal · Authors · 2024-08-07
>
> Thank you for your insightful comments and observations.
>
> ---
>
> **(W1)** We thank the reviewer for raising potential concerns regarding tokenizations. We already searched through different task designs to improve the performance of GPT-4 and Gemini-Pro. The reviews motivated us to conduct further investigations and quantification primarily on the openly available GPT-4 tokenizer. Our findings can be summarized as follows (see the general response for a detailed discussion):
>
> - We ablated different task designs in the initial submission; however, we found that the current sequence design (e.g., "ab1cd") performs best among other alternatives (such as "ab-1-cd").
> - As an additional analysis, we tested the tokenization of GPT-4's open-source tokenizer (tiktoken). We found that the digit delimiter is always tokenized separately (e.g., "ab1cd" yields "ab", "1", and "cd"); hence, it does not have a detrimental effect on the attention mechanism. We found that some of the 2-grams were split into 2 tokens; however, removing them from the dataset did not improve the overall task accuracy with the best-performing prompting technique.
> - We designed a new natural variation of PEN, dubbed Natural-PEN, where we replaced the synthetic 2-gram sequences with 3-gram, in-distribution, natural English words. Overall, we found that GPT-4's performance does not improve on this in-distribution, natural English words, yielding even a lower accuracy on Natural-PEN.
> ---
> **(W2)** Thanks for the comment and the relevant pointers! Following your suggestion, we provided the models with a more consistent number of shots on both PEN and PERM to verify whether increasing the number of examples beyond 8 would highlight an increase in the models' accuracy. We provided up to 32 examples (64 did not fit into the 8k context window of the GPT-4 version we are using). However, we did not observe any performance improvement, as reported in the table below.
>
> |Model|Task |Termination Acc.|Match Acc.|Task Acc.|
> |-----|---|----|----|----|
> |GTP-4|PEN|0.16|0.06|0.0|
> |Gemini-Pro|PEN|0.15|0.2|0.0|
> |GPT-4|PERM|0.36|0.59|0.0|
> |Gemini-Pro|PERM |0.32|0.05|0.0|
>
> We believe that the helpfulness of the additional examples might depend on the considered task and that, in some cases, there could be no improvements or even considerable sensitivity to the number of prompts. If you consider these results relevant, an Appendix on them could be added upon acceptance.
>
> ---
>
> **(W3)** This is a limitation of our current results. However, we did experiment with different sizes of LLaMA up to 600M. Because we could train a 33M model to perform the PEN task well, it is reasonable to assume a 150M model should be sufficiently large, and hence present our results with this model. Regarding other architectures, we also ablated a Universal-Transformer-style model in Appendix B.5 thanks to their good performance on compositional tasks like SCAN [61]. This model (UT-style LLaMA) has only one layer with the same configuration (embedding size 1024, 16 heads), and this layer is repeated. We achieved very similar results with UT-style LLaMA (Table B.10) compared to our main model. Please see our general response for more variants of architectures that we tried.
>
> ---
>
> **(Q1)** This is an interesting question, thanks for pointing to the prior work for discussion in our paper and suggesting a more general research direction. In our investigation, the tasks considered do not present unobserved local structures since the composition of primitives only admits a specific combination of operations (e.g., RIGHT[MATCH[LEFT[MATCH]]] in PEN), and the trained models should always have (at least in principle) the possibility to observe and learn this specific sub-graph in the computational graph.
>
> This is a desired property as the goal of our investigation was not to stress-test how well Transformer language models can learn arbitrary difficult compositional tasks but rather to measure how **sample efficient** they are in learning the composition of simple primitives in algorithmic tasks. Considering more tasks characterized by more complex phenomena (such as unobserved compositions of the primitives, spurious correlations in the data, semantical ambiguities, etc.) would have hence partially defeated this purpose, contaminating the effects of the intrinsic limitation of the Transformer model with other factors.
>
> However, after observing and validating our hypotheses on the simple synthetic datasets, we still believe that it would be compelling to extend the analysis to more natural compositional tasks such as the ones investigated by Bogin et al. It would allow us to verify how well the observed behavior scales in a real-world scenario, possibly providing new angles to it.
>
> ---
>
> **(Q2)** Please refer to the results presented in (W1) and to the note on tokenization included in the general response.
>
> ---
>
> **(Q3)** We report here the exact specifications of the models that we trained in our experiments. We use the Adam optimizer, a hidden size of 1024 and 12 layers. The model is a standard LLaMA 2 implementation, decoder only with Rope positional encoding and a Swiglu feedforward net. We did not use Grouped Query Attention, as LLaMA models do not use them in their smaller sizes.
> For what concerns the possible ablations on width/depth and larger models, we performed some minor ablations on them, but we observed no changes in the results, discouraging us from proceeding further in this direction (see our response in (W3) as well). However, as pointed out in your question, it would be interesting to observe whether the size and depth of the model can play a role in the learning efficiency, and if this interplay can be characterized through similar scaling laws such as Kaplan's.
>
> ---
>
> **(Q4)** We would like to extend our sincere gratitude for the suggestions on the presentation and writing in general. We will incorporate them upon acceptance.

---

> > ### Author Response · Authors · 2024-08-12
> >
> > Once again, thank you for your valuable inputs and service as a reviewer. We are at your disposal for any further clarification or discussion regarding our rebuttal.

---

> > > ### Comment · Reviewer_DGs4 · 2024-08-12
> > >
> > > Thank you for the detailed responses.
> > > Many of questions were responded to and clarified and I'm more confident in my evaluation.
> > > I would like to note that my main concern which is the synthetic nature of the experiments remains unanswered. I appreciate the authors attempt to provide additional experimental results in the short rebuttal time. However, using English-words as tokens does not make it "in-distribution" as these n-grams are completely out-of-distribution compared to the training data of the model. If anything, using common full-word tokens as inputs induce a strong prior as encoded in their embeddings and is thus expected to deteriorate the performance. It is not trivial to design a similar testbed that does simulate an in-distribution task, but is a pursuit that if achieved will be most beneficial. Nevertheless, even this intermediate step of investigating the compositional abilities of model in a synthetic controlled setting is interesting and enriches our knowledge about the properties of transformer based language models.
> > > I thus wish to thank the authors for the answers, and choose to keep my score as is.

---

> > > > ### Author Response · Authors · 2024-08-14
> > > >
> > > > Thanks for your additional comments on our rebuttal, we are glad to hear that many of your questions were clarified and that you are even more confident about your positive judgment on our submission.
> > > >
> > > > Regarding the synthetic nature of the experiments, we understand your concerns and we agree that the scope of the additional experiments on Natural-PEN might be limited if the goal is to evaluate to what extent the pre-training can improve the compositionality of these large models. This ablation was mostly proposed to check whether the use of "rare" tokens in the tasks had an impact or not on the performance.
> > > > We also agree that the design and evaluation of a similar testbed that simulates an in-distribution task will be valuable future work, but at the same time we still strongly argue for the importance of an evaluation on a synthetic setting which allows full controllability (see also the discussion on this with reviewers teuc and Fa35).
> > > >
> > > > Once again, thanks for your positive and constructive feedback and for your involvement in the review process, which we deeply appreciate.

---

### Official Review · Reviewer_qmuR · 2024-07-15

**Soundness:** 2
**Presentation:** 2
**Contribution:** 2
**Rating:** 4
**Confidence:** 2

**Summary:**

The paper investigates the capabilities of Transformer-based language models in learning compositional discrete tasks. The authors evaluate both training LLaMA models and prompting GPT-4 and Gemini-Pro on tasks that require the learning of compositions of several discrete sub-tasks. The results indicate that these models exhibit significant sample inefficiency: LLaMA models require more data to learn compositional tasks than to relearn all sub-tasks from scratch, and in-context prompting with GPT-4 and Gemini is unreliable and often fails in multi-round code generation. The findings are supported by a theoretical analysis showing the sample inefficiency of gradient descent in memorizing feedforward models.

**Strengths:**

- The paper evaluates both training from scratch and in-context prompting methods, providing a thorough analysis of the models' capabilities.
- The authors introduce new algorithmic tasks designed to test compositional learning and providing a theoretical framework to support the empirical findings.
- The study offers a deep dive into the limitations of current LLMs, supported by both empirical data and theoretical arguments, which can guide future research in learning compositional tasks.

**Weaknesses:**

- The tasks and settings used in the experiments may not cover the full range of real-world applications, limiting the generalizability of the findings.
- The performance and conclusions drawn are heavily dependent on the specific tasks designed by the authors, which might not fully represent other compositional learning scenarios.
- Personally, it took me a while to understand how the given algorithmic tasks are designed and how they relate to the broader context of compositional learning. For instance, the `PERM' problem was not immediately intuitive to me.

**Questions:**

- How well do the findings translate to practical, real-world applications beyond the synthetic tasks used in the experiments? Any specific reason to introduce new algorithmic tasks for evaluation?
- Would the models perform differently on a broader variety of compositional tasks, particularly those that are more complex or domain-specific?
- What specific modifications to model architecture or training strategies could be employed to enhance the sample efficiency of Transformer models in compositional learning?

**Limitations:**

- The study focuses on a limited set of compositional tasks, which may not fully capture the diversity of problems faced in real-world applications.

---

> ### Author Rebuttal · Authors · 2024-08-07
>
> Thank you for your insightful comments and questions.
>
> ---
>
> **(W1)** We acknowledge that the investigated collection of tasks does not cover the full range of real-world applications. Nonetheless, this is a common choice across different works in this domain [paper references 16, 29, 30]. It allows us to operate in a **fully controlled** environment, limiting the impact of exogenous factors on the empirical observations while having the possibility to stress-test compositionality. Furthermore, we believe that working with synthetic datasets does not limit the generalizability of our findings. When a model fails to learn efficiently in a synthetic, fully controlled environment, it will even more do so in more complex, real-world scenarios, which present several additional challenges for the model (e.g., semantic and syntactic ambiguities, spurious correlations, distribution shifts, etc.).
> This is also supported by a deep foundation of literature showing limited compositionality in practice [paper references 5, 8, 9, 11, 13 16].
>
> Moreover, the synthetic nature of the considered tasks does not preclude the possibility of making observations that could be translated to more complex, real-world scenarios as well. One example is the multiplication task (MUL), a fundamental task in mathematics on which LLMs struggle without a significant amount of additional training [paper references 11, 16]. We speculate that both our empirical and theoretical results can shed light on this behavior, justifying it as an intrinsic inefficiency of LLMs in composing the simple sub-operations necessary to solve the task and quantifying their sample inefficiency in certain complexity classes.
>
> ---
>
> **(W2)** While strictly speaking, our statements are limited to our tasks, in our investigation, we include a diverse range of algorithmic tasks and observe consistent behavior across all of them. Therefore, we expect that the same conclusions would hold on any similar tasks involving the composition of simple, known primitive operations (which occurs in many algorithmic, mathematical, and reasoning tasks).
>
> ---
>
> **(W3)** We tried to clarify the tasks with an updated Figure 1, shown in the 1-page PDF note.
>
> ---
>
> **(Q1)** Concerning the extent to which our observations could potentially extend to practical, real-world applications beyond the synthetic tasks used in the experiments, please refer to our observations in (W1). In addition, also refer to the general response (tokenization) with the additional results on the new Natural-PEN.
>
> On the other hand, the reasons behind the introduction of the novel tasks are manyfold:
> - First, we argue for the introduction of algorithmic tasks based on the pointer execution, introduced by Bengio et al. [20] and so far absent in the landscape of research on compositional learning. This task limits the number of confounders in the data and, therefore, reduces (if not annihilates) the number of shortcut solutions that the model can find. In other words, pointer execution tasks force the model to learn an algorithmic (general) solution by design and not rely on any possible statistical shortcut that might compromise the validity of the observations.
> - As algorithmic tasks, PEN and PERM are particularly suitable for benchmarking compositionality. We can decompose them into single atomic operations (referred to as _primitives_ in the text), which can make learning the task easier if the model can identify and leverage their compositional nature. We clarify their decompositional nature in the response of W2 of reviewer Fa35.
> - Finally, our tasks represent a unique example in the sense that the failure of GPT-4 and Gemini-Pro is very apparent despite extreme help, e.g., providing multi-shot examples and showcasing limited compositionality of current SOTA models better than any task we are aware of.
>
> ---
>
> **(Q2)** These LLMs would always perform decently if there is enough data. Our findings, however, are that the required data becomes excessive with more complex tasks. Our forecast is, therefore, that the more complex and the more domain-specific the tasks become, the harder it becomes for LLMs to perform them correctly (e.g., different to our education systems, where a student does not necessarily get more data for more complicated concepts). Concretely, in our experiments, this materializes in a model starting to hallucinate wrong solutions and failing at seemingly obvious executions (see GPT-4 experiments and Figure 2).
>
> ---
>
> **(Q3)** There exists evidence [17, 61] that Transformers with weight-sharing between different layers and recurrence tend to perform better on compositional tasks than standard Transformer variants. It is for this reason that we also experimented with several such models during our work. We discuss details thereof in the general response.
>
> Additionally, we independently investigated several novel architectural modifications aimed at increasing compositionality within the Transformer. One such idea has been penalizing superposition in favor of improving compositionality inspired by [Elhage et al., 2022, Olah, 2023], by including a single parameter modifying the read-outs from the latent representations for attention and feedforward blocks that penalize superposition and with that possibly encourages compositionality in the latent representations. We also built a Transformer that appends new computations from attention and feedforward passes (instead of adding) to a list of "latent tokens" with an additional attention mechanism to access those fields in the hope that compositionality can be encouraged. While some of those approaches did increase compositional capabilities on small datasets, we have not found a fundamental improvement in the domain of our experiments so far.
>
> [Elhage et al., 2022] https://arxiv.org/abs/2209.10652
> [Olah, 2023] https://transformer-circuits.pub/2023/superposition-composition/index.html

---

> > ### Author Response · Authors · 2024-08-12
> >
> > Once again, thank you for your valuable inputs and service as a reviewer. We are at your disposal for any further clarification or discussion regarding our rebuttal.

---

### Author Rebuttal · Authors · 2024-08-07

We want to thank all the reviewers for their helpful and supporting comments. We are encouraged that they acknowledge the importance of addressing current Transformer models'  limitation in learning compositional tasks (DGs4, teuc), and that w.r.t. other benchmarks such as BIG-bench, our work provides a concrete, quantitative, and extensive study on the extent of this insufficiency (teuc) by introducing two novel evaluation benchmarks (i.e., PEN and PERM). Moreover, we are glad that they appreciate the thoroughness of our analysis (teuc, qmuR) with both from scratch and in-context learning, as well as theoretical proofs (teuc, Fa35, qmuR).

In this general response, we summarize our response to the reviewer's major concerns. A point-by-point response to each reviewer can be found in the individual responses.

## Tokenization
We thank the reviewers for raising potential concerns regarding tokenizations. Indeed, we already searched through different task designs to improve the performance of GPT-4 and Gemini-Pro. The reviews motivated us to conduct further investigations and quantifications primarily on the openly available GPT-4 tokenizer. Our findings can be summarized as follows:

**Different task designs**
While preparing the initial submission, we already ablated the structure of the PEN and PERM examples fed into the LLMs to validate whether differences in the representation of the task could affect the tokenization and, indirectly, the performance of the model. One example of an ablated structure involved adding dashes to the sequences (e.g., "ab-1-cd"). However, these variations in the structure did not achieve any improvement over our base formulation of the tasks.

**Tokenizer analysis with GPT-4**
To shed more light on tokenization, we additionally tested the validity of the prompting technique reported in the manuscript for GPT-4 using its open-source tokenizer (tiktoken). Unfortunately, we could not find an open-source tokenizer for Gemini-Pro; the available token counter only provides limited information about the actual tokenization. Analyzing GPT-4's tokenization on our tasks showed that our induced sequence structure with digits as delimiters ensures safe splitting within a word, e.g., "ab1cd" would be mapped to three tokens "ab", "2", and "cd". However, we saw that 13.2% of the 2-grams are split into two tokens (e.g., "bq" is tokenized to "b" and "q"), which may affect the attention mechanism. As an additional experiment, we removed these 2-grams from the dataset. However, this setup could not improve the performance (0.05 vs. 0.19 on PEN) when using the best-performing prompting technique with GPT-4 (Code Interpreter). Hence, tailoring the task to the tokenizer does not lead to an improved performance.

**New Natural-PEN with in-distribution, natural English words**
Inspired by the comments of reviewer Fa35, we additionally designed a natural variation of the PEN, dubbed Natural-PEN, where we replaced the synthetic 2-gram sequences used for the matching in the original tasks with 3-gram, in-distribution, natural English words. To do so, we filtered all valid 3-gram words from Scrabble (https://scrabble.collinsdictionary.com/word-lists/three-letter-words-in-scrabble/) that resulted in a single token when using GPT-4's tokenizer. This gave us a set of 707 words (out of the original 1338 words). Similar to the experiment in the previous paragraph, we find that the GPT-4's performance using the best prompting techniques does not improve on this in-distribution, natural English words, yielding even a lower accuracy on Natural-PEN:

|Setup |Termination Acc.|Match Acc.|Task Acc.|
|-----|----|----|----|
|Few-shot CoT|0.5|0.27|0.0|
|Few-shot CoT, traps removed|0.2|0.29|0.0|
|Code Interpreter|0.05|0.1|0.05|


## Many-shot prompting of GPT-4 and Gemini-Pro
Following suggestions of DGs4, we increased the number of shots from 8 examples up to 32 examples (64 did not fit into the 8k context window of the old GPT-4 version we are benchmarking). However, we did not observe any performance improvement, as reported below.

|Model|Task |Termination Acc.|Match Acc.|Task Acc.|
|-----|---|----|----|----|
|GTP-4|PEN|0.16|0.06|0.0|
|Gemini-Pro|PEN|0.15|0.2|0.0|
|GPT-4|PERM|0.36|0.59|0.0|
|Gemini-Pro|PERM |0.32|0.05|0.0|

We believe that the helpfulness of the additional examples might be dependent on the considered task and that, in some cases, there could be no improvements or even considerable sensitivity to the number of prompts.

## Different architectures/configurations
Several reviewers rightfully questioned whether the architecture we used should be expected to learn compositionally. We obtained results with several different architectures based on the Universal Transformer (UT) [61] and the more recent Hyper-UT [17], for both of which there is evidence that they exhibit better compositional generalization than the original Transformer. For instance, a UT-style transformer displayed good performance on compositional tasks like SCAN [61]. The model based on UT reached a similar performance as the original LLaMA model, as documented in Appendix B.5, which placed it into the H4 category. We then also experimented with a Hyper-UT-style LLaMA, based on the aforementioned Hyper-UT work [17], in which each layer selects its weights from a common weight embedding pool to realize a parameter-efficient model. This model consistently achieved accuracies below that of LLaMA on a large set of small algorithmic tasks. Thus, we chose not to systematically explore its performance in a more challenging compositional algorithmic setting. However, to improve the generalizability of our findings, we will include results with this particular architecture in a new Appendix.

---

### Decision · Program_Chairs · 2024-09-25

**Decision:**

Accept (poster)

**Comment:**

This paper presents empirical and theoretical evidence that Transformer language models are inefficient learners of compositions of tasks, in the sense that they require more training samples than the sum of the samples required to learn each task individually. The reviewers identified several strengths, including
1. Well-designed problems that are carefully constructed to shed light on an important question
2. Through empirical experiments using both open-weight and API-access models
3. Clear writing and a very thorough literature review that places the results in context

Two of the reviewers raised two significant concerns. First, qmuR and Fa35 felt that the contrived tasks were not realistic. As the authors replied in their rebuttal, real world data introduces many confounders and controlled experiments are valuable for understanding the limitations of a model. Further, the authors emphasized that the "algorithmic" nature, i.e., having a precise, formal specification improves on previous work studying composition, and extends prior work using the pointer-based tasks that did not explore composition. The AC agrees with the authors on these points.

The second major concern was about experimental details, including concerns about tokenization as a confounder. The authors addressed these concerns in their rebuttal and they are encouraged to include the additional investigations in the final version of the paper.